# When Consensus Is Not Correctness: Diversity Collapse and Manufactured Overconfidence in Multi-Agent LLM Debate

## Abstract

Multi-agent large language model (LLM) debate is widely believed to improve answers, and agreement is routinely read as evidence of trustworthiness. We show that debate transforms agreement from evidence into an outcome: agreement is *endogenous* to the interaction that produces it. A variance account, tied to measured diversity collapse, makes this precise. As agents read one another, the inter-agent correlation rises toward one. That same correlation controls both the panel's error and the disagreement an operator reads as confidence, driving them in opposite directions: the ensemble stops averaging out error exactly as it stops looking uncertain. Three consequences follow. Apparent confidence saturates independently of error. The central empirical signature is not the identity $G = C - A = R - (1 - C)$, but the collapse-induced flattening of the confidence shortfall $1 - C$: terminal confidence has $17\times$ smaller variance than accuracy, and one shared shortfall predicts per-condition gaps out of sample. Consistent with this signature, the induced gap–residual regression is affine on the primary model (slope 0.82, $R^2 = 0.96$) and remains monotone with sub-unit slopes across model-family probes. Whether debate is benign is then a race between error correction and confidence inflation, governed by role design and task headroom. We introduce *Calibrated Multi-Agent Debate*, a certification-first framework with two conditional levers, Prevent and Detect, and a split-conformal certificate, Certify. With exchangeable labeled calibration, Certify controls set coverage under collapse, at the cost of larger sets or abstention on hard cases, while agreement-based stopping commits confident errors at 18–47% miscoverage. Matched self-critique and verdict-injection controls separate interaction-driven amplification from baseline model overconfidence.

## 1 Introduction

Multi-agent large language model (LLM) debate (Du et al., 2023; Li et al., 2023; Hong et al., 2023) runs several model instances over multiple turns and aggregates their answers. It is increasingly embedded in agentic pipelines where the panel's apparent confidence decides whether a system answers, abstains, or escalates. A pervasive intuition, inherited from human deliberation and from ensemble learning (Hansen & Salamon, 1990; Krogh & Vedelsby, 1995; Dietterich, 2000; Lakshminarayanan et al., 2017), is that agreement among agents makes an answer more likely to be correct. Agreement is then used both as a confidence signal and as a stopping criterion: debate until consensus. The same equation of consensus with confidence runs well beyond debate, underwriting self-consistency decoding, majority voting, and the use of one model to judge another. This paper argues, theoretically and empirically, that this intuition is misleading in an important and predictable regime. Debate does reliably produce agreement, but that agreement is *endogenous* to the protocol, manufactured by the very interaction whose output it is taken to certify, so it cannot serve as a calibrated correctness signal. Read as confidence, it amplifies overconfidence through the debate interaction (Figure 1). The implication is that *self-produced agreement cannot certify its producer*. Agreement is a channel for proposing answers, not for certifying them.

Consider three agents answering a hard question. After one or two turns they often converge on a shared answer and report high confidence. The tempting response is to trust the panel. An operator, or an

automated gate, sees a confident and unanimous verdict and commits to the answer. But unanimity among models that have read each other's arguments is weak evidence, since the agents may have assimilated rather than verified. What makes this dangerous is not the error itself but the confidence attached to it: an answer marked uncertain invites a second look, whereas a confidently wrong consensus passes silently through the very gates meant to catch it. We show that on tasks where the model is fundamentally uncertain the agents converge anyway, to a confident wrong consensus. The signal an operator most wants to trust is thus least reliable exactly where the task is hardest.

The mechanism is a variance effect made precise by a classical identity. Treat each agent's turn-$t$ output as a noisy correctness estimate $X_i(t)$ with per-agent variance $\sigma^2(t)$ and inter-agent correlation $\rho(t)$. The team's aggregate obeys the classical equicorrelation identity (Cochran, 1977)

$$\mathrm{Var}(\hat{\mu}_t) \;=\; \frac{\sigma^2(t)}{N}\big[1 + (N-1)\rho(t)\big]. \tag{1}$$

Reading each other's outputs makes the agents assimilate, a consensus dynamic long studied in networked multi-agent systems (Olfati-Saber et al., 2007). The correlation $\rho(t)$ rises toward 1, producing a *diversity collapse*. As $\rho(t) \to 1$, the bracket $\to N$ and the $1/N$ averaging benefit vanishes: the ensemble no longer reduces error, even though the observed disagreement among agents goes to zero. This is the same disagreement an external reader, and often the agents themselves, use to gauge certainty. Apparent certainty thus rises while real error does not. Aggregation helps only when members err independently, and debate is the procedure that dissolves that independence, so the very interaction meant to improve the answer removes the condition under which pooling answers reduces error. A single correlation $\rho(t)$ drives both quantities, in opposite directions: the ensemble literature (Kuncheva & Whitaker, 2003; Brown et al., 2005) reads $\rho$ only through aggregate accuracy, but the same $\rho$ also governs the dispersion an operator reads as confidence, so as it climbs the panel looks most unanimous exactly as its averaging stops working. We call this gap *manufactured overconfidence*. Made quantitative, the prediction is sharp: the gap equals the model's residual error up to the measured confidence shortfall and tracks it monotonically, so the overconfidence is largest exactly where the model is least accurate. Debate is not uniformly harmful, though: error-correction can lower the residual even as confidence inflates, so whether a panel ends calibrated or overconfident is a race between the two, set by the roles the agents play and the headroom the task leaves.

Inter-agent agreement is therefore not a safe correctness signal. Interaction has turned a putative evidence source into an outcome of the protocol. The residual-tracking law above is what makes this frame testable rather than only conceptual. Parts of the resulting overconfidence can be prevented or detected in the regimes where their signals apply, but the stance must be *certification-first*. An endogenous signal cannot certify itself; safety must come from a quantity computed outside the collapsing interaction.

**We make three key contributions:**

▶ **Manufactured-overconfidence discovery.** We show that multi-agent debate manufactures additional overconfidence: terminal confidence loses dynamic range while accuracy varies, and one shared shortfall predicts gaps out of sample. The resulting gap tracks residual error and is worst where the model is least accurate. This holds across fifteen tasks, two roles, five model families, and free-form generation, and is not cured by adding agents; `GPT-4o-mini` is the weakest and least-saturated boundary case. A matched self-critique control and counterfactual verdict-injection probe separate interaction-driven assimilation from baseline model overconfidence.

▶ **A variance account of diversity collapse.** We trace it to a *diversity collapse*: as agents read one another the inter-agent correlation rises toward one, driving true error and observed agreement in opposite directions. Confidence then saturates independently of the error, the gap tracks the residual, and a role-and-headroom race sets the calibrated-versus-overconfident outcome (Theorems 1, 2, 3); terminal confidence loses its dynamic range (Corollary 1), while one shared constant predicts each condition's gap out of sample on the saturated families as shown in Section 4 and Appendix B.4.

▶ **A certification-first framework.** We introduce Calibrated Multi-Agent Debate (CMAD), an endogeneity-to-exogeneity ladder. *Prevent* hides peers' verdicts; *Detect* scores the consensus tra-

jectory for selective prediction (Geifman & El-Yaniv, 2017); *Certify* replaces agreement with a split-conformal set. Only Certify is collapse-unconditional: with an exchangeable labeled calibration split, it holds set coverage $\geq 1 - \alpha$ even under collapse. Prevent and Detect are conditional, model-dependent levers. Section 5 and Appendix C give the formal framework and variants.

The variance identity equation 1 is classical and split-conformal prediction (Vovk et al., 2005; Angelopoulos & Bates, 2021) is established; what is new is the overconfidence account obtained when equation 1 is read under measured collapse, its quantitative match to data, the trajectory-trust corollary that operationalizes Detect, and Affirm's coupling of conformal coverage with a collapse-aware stopping rule.

## 2 Related Work

**Multi-agent debate and the ensemble-variance lens.** Du et al. (2023) introduced iterative multi-agent debate, and frameworks such as CAMEL (Li et al., 2023), MetaGPT (Hong et al., 2023), and ChatEval (Chan et al., 2024) scaled role-structured collaboration. Later work tempered this optimism. Wynn et al. (2025) and Smit et al. (2024) find regimes where debate does not reliably beat a single model or self-consistency; Kong et al. (2026) show that homogeneous agents converge toward near-identical outputs, a *semantic collapse*, over repeated interaction; Liang et al. (2024) document a *degeneration of thought* in which an agent, once confident, stops producing novel arguments; and Yang et al. (2026) argue, information-theoretically, that output diversity, not agent or turn count, is what sustains gains. These observations share a mechanism the classical ensemble-variance literature already names: the equicorrelation identity equation 1 is standard (Cochran, 1977), as is the bias–variance–covariance decomposition of ensemble error (Ueda & Nakano, 1996), with correlated members weakening the variance-reduction benefit of averaging. We make this lens central for debate in two steps: by treating the correlation as time-varying and assimilative, so "diversity collapse" is exactly $\rho(t) \to 1$, and, crucially, by linking the same $\rho(t)$ that governs $\text{Var}(\hat{\mu}_t)$ to the score dispersion $\mathbb{E}[S^2(t)]$ (Lemma 1). As this dispersion collapses, the panel appears more confident to an operator. The opposite limits of aggregate variance and apparent agreement as $\rho \to 1$ are the source of the overconfidence, and we trace the consequence not for accuracy but for calibration, predicting a residual-tracking signature whose magnitude we validate empirically (Theorem 2).

**Confidence, calibration, and agreement as a signal.** Language models are frequently miscalibrated and overconfident (Guo et al., 2017; Minderer et al., 2021; Ovadia et al., 2019; Tian et al., 2023; Kuhn et al., 2023; Xiong et al., 2024), and recent work surveys and empirically benchmarks the confidence-estimation and uncertainty-quantification methods proposed in response (Geng et al., 2024; Huang et al., 2025). Against this backdrop, a widespread heuristic treats inter-sample or inter-agent agreement as a confidence proxy in self-consistency, majority vote, and debate-until-consensus protocols (Wang et al., 2023). The assumption is that agreement tracks correctness. The same self-produced-signal assumption underlies LLM-as-judge evaluation (Zheng et al., 2023) and self-verification, where, consistent with our diagnosis, LLMs are found unable to reliably self-correct reasoning without external feedback (Huang et al., 2024). That assumption is exactly what our analysis overturns (Theorems 1–2): under diversity collapse the agreement proxy is not just noisy but systematically overconfident; when reported confidence co-saturates with that proxy, the observed bias tracks the residual error, so it is least reliable on exactly the hardest inputs, where it is most relied upon.

**Stopping rules and conformal certification.** A line of work decides when to stop debating: Hu et al. (2025) detect distributional stability of the answer sequence; Chang & Chang (2025) provide budget-aware termination with multiple gating signals; Eo et al. (2025) debate only when single-model confidence is low; and Fan et al. (2025) learn a binary trigger for when debate is worthwhile. These rules monitor some form of convergence or self-reported confidence, the very signals our theory flags as corrupted. Conformal prediction offers an exogenous alternative: split conformal gives distribution-free, finite-sample coverage under exchangeability (Vovk et al., 2005; Angelopoulos & Bates, 2021), with adaptive multiclass set constructions (Romano et al., 2020; Angelopoulos et al., 2021), anytime-valid extensions via test martingales (Ramdas et al., 2023), and weighted variants under covariate shift (Tibshirani et al., 2019). In the LLM setting, Wang et al. (2026) attach a conformal layer as a post-hoc act-versus-escalate gate and Zhou et al. (2026)

allocate error budgets across turns. We put the same coverage machinery to a different conceptual use: an explicit replacement for the agreement signal the theory shows is unsafe, coupled to a collapse-aware stopping rule whose marginal coverage provably survives diversity collapse (Theorems 4–5). Affirm thus stops on a coverage-validated quantity, the set size, rather than on convergence, with a strict, quantified safety separation from agreement-based stopping (Corollary 3).

## 3 Preliminaries

A debate runs $N$ agents with fixed role prompts over $T$ turns. At turn $t$, agent $i$ conditions on the transcript through turn $t-1$ and emits an answer $\hat{y}_i(t) \in \mathcal{Y}$ and a self-reported confidence $c_i(t) \in [0,1]$. For multiple choice, $\mathcal{Y} = \{1, \ldots, K\}$; the gold label is $y$. Instances are i.i.d. from a distribution $\mathcal{P}$. All per-turn quantities below are population objects estimated by Monte Carlo in Section 7. Exchangeable agents justify the single scalar equicorrelation $\rho(t)$; the heterogeneous-covariance extension is deferred to Appendix B.10. Appendix A collects the notation used throughout the paper.

**Definition 1** (Correctness score and observable agreement). For the variance analysis we use a per-agent *correctness score* $X_i(t) = \phi(\hat{y}_i(t), c_i(t)) \in [0,1]$. It equals $c_i(t)$ if $\hat{y}_i(t) = y$ and $1 - c_i(t)$ otherwise. This score needs the gold label and enters only the proofs. Let $\sigma^2(t) = \text{Var}(X_i(t))$, equicorrelation $\rho(t) = \text{Corr}(X_i(t), X_j(t))$, aggregate $\hat{\mu}_t = \frac{1}{N} \sum_i X_i(t)$, and dispersion $S^2(t) = \frac{1}{N} \sum_i (X_i(t) - \hat{\mu}_t)^2$. Two *confidence* signals run alongside. The agents *report* confidence directly: the panel's *reported confidence* $C(t) = \frac{1}{N} \sum_i c_i(t)$ is the mean verbalized confidence, and is the quantity that enters the calibration gap and every empirical measurement in Section 7. What an operator instead reads at deployment is *label-free*: an *apparent confidence* $\widetilde{C}(t)$, a continuous statistic of the agents' observed agreement on answers and confidences. One example is the *answer consensus* $\kappa(t)$, the fraction of votes on the modal answer. Apparent confidence is maximal at unanimity, where $\widetilde{C} = 1$. Under collapse, both $S^2(t)$ and the observed disagreement vanish, so the saturation we prove via $S^2$ holds for the apparent confidence $\widetilde{C}(t)$; the reported confidence $C(t)$ is measured separately and empirically co-saturates in our debates (Section 7).

**Lemma 1** (The collapse-dynamics identity). *Under Definition 1 with equicorrelation $\rho(t)$,*

$$\text{Var}(\hat{\mu}_t) = \frac{\sigma^2(t)}{N} \big[1 + (N-1)\rho(t)\big], \qquad \mathbb{E}\big[S^2(t)\big] = \frac{N-1}{N} \sigma^2(t) \big(1 - \rho(t)\big). \tag{2}$$

As $\rho(t) \to 1$, these quantities move in opposite directions. The aggregate variance approaches $\sigma^2(t)$, so the averaging benefit is lost, while $\mathbb{E}[S^2(t)] \to 0$, so the agents look unanimous. That divergence is the source of the manufactured overconfidence. Proof in Appendix B.1.

**Assumption 1** (Diversity collapse). Debate is assimilative: $\rho(t)$ rises over the horizon toward a limit $\rho_\infty$, the per-agent variance is bounded ($0 < \sigma^2(t) \leq \sigma_{\max}^2$), and both the latent dispersion $\mathbb{E}[S^2(t)]$ and the agents' observed disagreement on answers and confidences decrease over the horizon. Equivalently, apparent confidence rises. In the collapse limit, observed disagreement vanishes and apparent confidence approaches one as $\rho_\infty \to 1$; the debate *collapses* if $\rho_\infty = 1$. The equicorrelation form is a convenience, not essential: Appendix B.10 shows the same collapse and all of the apparent-confidence conclusions hold for an arbitrary inter-agent covariance; the reported-confidence gap still uses the empirical co-saturation condition stated in Theorem 2. We verify $\rho_\infty \approx 1$ empirically across all conditions (Section 7, Figure 2).

## 4 A Variance Theory of Debate Overconfidence

We track the *accuracy* $A(t) = \mathbb{P}(\hat{y}^{\text{agg}}(t) = y)$ of the aggregated decision. Here $\hat{y}^{\text{agg}}(t)$ is the modal answer among the agents' $\hat{y}_i(t)$. We also track the two confidence signals of Definition 1: apparent confidence $\widetilde{C}(t)$, which the variance mechanism drives, and reported confidence $C(t)$, which the experiments measure. The *loss* $\ell(t) = 1 - A(t)$ decomposes into a reducible part, such as noise that debate can average down or errors that critique can fix, and an *irreducible residual* $R_\infty = \lim_t \ell(t)$, the error the debate process cannot remove, such as a shared misconception or genuine task ambiguity. The *calibration gap* $G(t) = C(t) - A(t)$ is the amount by which the reported confidence exceeds accuracy; its label-free analogue $\widetilde{C}(t) - A(t)$ is what an operator reading agreement would see.

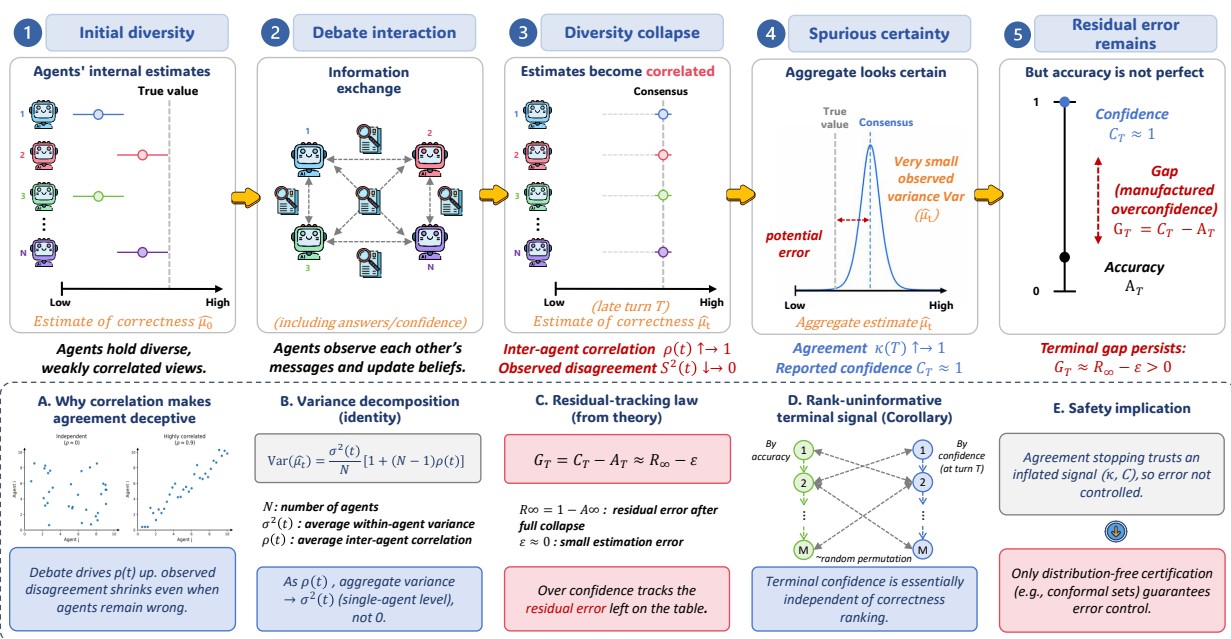

Figure 1: **How Debate Turns Agreement into Manufactured Overconfidence.** Debate increases inter-agent correlation, causing observed disagreement to collapse and agreement-based confidence to saturate. Because residual error can remain after collapse, the terminal calibration gap tracks the residual error rather than correctness.

## 4.1 Confidence saturates independently of correctness

Collapse drives the agents' observed disagreement to zero, so any confidence read from it rises to its ceiling regardless of the panel's accuracy.

**Theorem 1** (Confidence saturation under collapse). *Under Assumption 1 with $\rho_\infty = 1$ and $\sigma^2(t) \leq \sigma_{\max}^2 < \infty$, the dispersion vanishes, $\mathbb{E}[S^2(t)] \to 0$; the observed disagreement also vanishes in this collapse limit by Assumption 1, so the apparent (agreement-based) confidence $\widetilde{C}(t)$ of Definition 1 satisfies $\widetilde{C}(t) \to 1$. The limit is independent of correctness: it is the same whether the residual error $R_\infty$ is zero or close to one.*

The proof (Appendix B.2) is immediate from Lemma 1: $\mathbb{E}[S^2(t)] = \frac{N-1}{N}\sigma^2(t)(1 - \rho(t)) \to 0$, and by Assumption 1 the observed agreement saturates as $\rho(t) \to 1$, so continuity at unanimity gives $\widetilde{C}(t) \to 1$. The signal a reader uses to judge certainty is thus driven entirely by $\rho(t)$, which debate inflates, so the saturated signal cannot identify the bias.

*Reported confidence co-saturates empirically.* The theorem is a statement about the *apparent* confidence $\widetilde{C}(t)$, a label-free statistic of agreement, and nothing more: it does *not* assume that the agents' verbalized confidence is a deterministic function of agreement, only that the agreement-based signal collapses. The *reported* confidence $C(t) = \frac{1}{N}\sum_i c_i(t)$ the experiments measure is therefore a separate quantity, not forced by the variance identity. We therefore treat co-saturation as an empirical prediction: reported confidence should rise with the apparent signal and approach a ceiling $C_\infty = 1 - \varepsilon$ with a small, near-constant shortfall $\varepsilon \approx 0.037$. This co-saturation is an empirical regularity, established by two independent measurements in Section 7. The across-condition variance of $C_T$ is $17\times$ smaller than that of accuracy, and a *single* shared shortfall predicts every condition's gap out of sample. Most directly, the reported shortfall $1 - C(t)$ contracts turn-by-turn together with the dispersion $\mathbb{E}[S^2(t)]$ the theorem acts on (Figure 4: Pearson $r = 0.60$ with bootstrap CI $[0.48, 0.70]$, Spearman $\rho = 0.54$, over all condition–turn pairs; both signals reach the same terminal tier), so the verbalized signal is empirically linked to exactly the quantity the collapse drives to zero. The calibration gaps below are stated for this reported $C(t)$, the operative empirical quantity; the

residual-tracking it exhibits is thus an empirical *signature* resting on the *proven* apparent-confidence collapse plus this *verified* co-saturation, not a property claimed for verbalized confidence by the theory alone.

**Corollary 1** (Terminal confidence loses its dynamic range under co-saturation)**.** *Fix a task. For the apparent confidence this follows from Theorem 1; for the reported confidence, assume the empirical co-saturation described above. Then terminal confidence approaches a common ceiling on every instance of that task, so its across-instance variance vanishes, $\mathrm{Var}_x[C_\infty(x)] \to 0$, and its usable dynamic range collapses. At exact saturation, $C_\infty$ is constant in $x$. Every correct and incorrect pair then ties, so the correctness-ranking area under the ROC curve (AUROC) is exactly $\frac{1}{2}$; near collapse any residual discrimination survives only in the vanishing shortfall $\varepsilon(x) = 1 - C_\infty(x)$, which collapse does not by itself drive to chance. A label-free signal that reliably discriminates must instead come from the trajectory before collapse, for instance the rate at which consensus formed, where $\rho(t) < 1$ and the dispersion still carries information.*

Corollary 1 grounds the *Detect* component in Section 5.2. The usable confidence signal loses its dynamic range precisely at the turn an operator reads it, so any remaining rank signal can only live in the vanishing shortfall. On hard tasks this shortfall is empirically near chance, with within-task AUROC 0.49–0.58 in Section 7. The surviving signal is instead how the consensus was reached, where consensus speed retains discrimination. Proof in Appendix B.3.

## 4.2 The overconfidence gap tracks the residual error

Once confidence has saturated, the calibration gap can no longer reflect how often the panel is right; what remains is the one error collapse cannot average away, the irreducible residual.

**Theorem 2** (Overconfidence tracks the residual error)**.** *Let $C_\infty = \lim_t C(t)$ and $A_\infty = \lim_t A(t) = 1 - R_\infty$. Then the terminal calibration gap is*

$$G_\infty = C_\infty - A_\infty = R_\infty - (1 - C_\infty). \tag{3}$$

*If, in the collapse limit, the reported confidence co-saturates with the apparent signal ($C_\infty \to 1$), then $G_\infty = R_\infty$: the manufactured overconfidence then equals the model's residual error. More generally, if confidence saturates to $C_\infty = 1 - \varepsilon$ (with shortfall $\varepsilon \geq 0$ possibly depending weakly on the task), then $G_\infty = R_\infty - \varepsilon$. For two tasks with residuals $R_1 < R_2$ and shortfalls $\varepsilon_1, \varepsilon_2$, the gap increases whenever $0 \leq \varepsilon_2 - \varepsilon_1 < R_2 - R_1$. Thus, if the shortfall grows more slowly than the residual over the task family, $G_\infty$ tracks $R_\infty$ monotonically. In the smooth or affine case the local slope is $1 - \mathrm{d}\varepsilon/\mathrm{d}R_\infty \in (0, 1]$. Constant $\varepsilon$ gives unit slope; the observed slope is an empirical quantity measured in Section 7.*

The identity alone is not the claim. Without confidence saturation, the shortfall $1 - C_\infty$ could vary arbitrarily with task difficulty and destroy any residual-tracking prediction. The testable content is that debate collapses this shortfall to a nearly constant level, so the algebraic identity becomes predictive across tasks and models.

**Corollary 2** (Where debate is most dangerous)**.** *Under the monotonic-shortfall condition of Theorem 2 ($0 \leq \varepsilon_2 - \varepsilon_1 < R_2 - R_1$ whenever $R_1 < R_2$), $G_\infty$ is increasing in $R_\infty$: debate manufactures the most overconfidence precisely on the tasks where the model is least able to be right (hard or ambiguous tasks with large irreducible error) and essentially none on saturated tasks ($R_\infty \approx 0$), for any role design that reaches this collapse regime.*

The prediction is sharply falsifiable: plotting the terminal gap against the residual error $1 - A(T)$ across tasks should yield a relation that increases monotonically with the residual, with local slope in $(0, 1]$. Unit slope appears only when the shortfall $\varepsilon$ is constant under full saturation. The theory commits only to this monotone, residual-tracking shape, not to any functional form or slope; that the relation is empirically affine with a *sub-unit* slope (Section 7 measures 0.82, $R^2 = 0.96$) is an empirical signature, not predicted. The identity $G_\infty = R_\infty - \varepsilon$, with $\varepsilon = 1 - C_\infty$, is algebraic; its empirical content is that the measured shortfall $\varepsilon$ collapses to a near-constant, paralleling the apparent-confidence collapse in Theorem 1. That co-saturation is what pins the gap to the residual. Proof in Appendix B.4. A finite-time refinement (Proposition 1, Appendix B.5) bounds the *apparent* gap at every turn by the measurable collapse factor $1 - \rho(t)$, so the residual-tracking holds at the turn an operator reads the panel, not only in the limit.

### 4.3 Whether debate is benign depends on roles and task headroom

Within the collapse regime, what distinguishes benign debate from harmful debate is not whether agents converge but whether accuracy keeps up with the rising confidence.

**Theorem 3** (Productive–overconfident dichotomy). *Over a horizon $[1, T]$ write $\Delta C = C(T) - C(1)$ and $\Delta A = A(T) - A(1)$. When the reported confidence inflates over the collapse horizon, $\Delta C \geq 0$. The change in the calibration gap is $\Delta G = \Delta C - \Delta A$, so*

$$\Delta G < 0 \ (gap\ decreases) \iff \Delta A > \Delta C \geq 0, \tag{4}$$

*and the overconfidence gap decreases exactly when error-correction $\Delta A$ outpaces the (nonnegative) confidence inflation $\Delta C$. In regimes with positive confidence inflation, the sign of $\Delta G$ is governed by the error-correction rate $\Delta A$, which depends on role design and task headroom.*

The criterion $\Delta A$ versus $\Delta C$ predicts a negative association between accuracy gain and gap change; in the overconfident regimes studied here, the analogous expected calibration error (ECE) pattern is observed as well (Section 7 finds $r(\Delta A, \Delta \text{ECE}) = -0.46$). Three regimes follow: cooperative roles on tasks with headroom ($\Delta A$ large, productive); adversarial or persuasion roles ($\Delta A \approx 0$, overconfident); and ambiguous tasks where no role can raise accuracy ($\Delta A \approx 0$ for both, overconfident regardless of role, the maximal-danger case of Corollary 2). Proof in Appendix B.6.

## 5 Calibrated Multi-Agent Debate Framework

The variance account diagnoses an *endogenous* trust signal (Section 4), corrupted by collapse in proportion to how close it sits to the interaction it certifies. *Calibrated Multi-Agent Debate* (CMAD) responds with three components that trust this signal progressively less: *Prevent* reduces the endogeneity at the protocol level, *Detect* reads it from the trajectory before it saturates, and *Certify* replaces it with an exogenous coverage guarantee. This account motivates routing between them: role design and headroom (Theorem 3) identify when Prevent can help, Corollary 1 identifies when Detect is needed, and the certificate holds under exchangeability even when collapse occurs (Theorem 4). Only the exogenous layer is collapse-unconditional: the two signal-based components are model-dependent (Section 7), and Proposition 3 shows they can change only efficiency, not the coverage guarantee under the stated exchangeability conditions. CMAD is therefore intentionally *asymmetric*, a routing framework rather than a single always-on intervention. *Prevent* is a protocol regularizer for suspected role or channel coupling. *Detect* is gated by terminal-confidence dynamic range, as evaluated in Section 7. *Certify* is the coverage-bearing component and is reported by default whenever a guarantee is needed. Prevent and Detect are efficiency and routing levers, not universal interventions; only Certify carries the coverage guarantee.

### 5.1 Prevent: Verdict-Sequestered Debate

Prevent acts at the protocol level, on the most salient channel by which agents condition on one another's answers.

**Definition 2** (Verdict-Sequestered Debate (Sequester)). Standard debate shows each agent the full text of its peers' previous turns, including the committed line `ANSWER: k; CONFIDENCE: c`. *Verdict-Sequestered Debate* is the identical protocol with one change: before a peer's message is shown to others, the committed answer/confidence line is withheld (replaced by a redaction token), so agents exchange reasoning but not their stated answers or confidences. Every agent still emits its own answer and confidence for scoring; only the peer-visible copy is redacted. Sequester is a one-line change to the message relay and adds no cost.

Sequester withholds one salient route by which agents condition on each other: the visible verdict. It targets part of the coupling that drives $\rho(t) \to 1$ (Lemma 1), but leaves reasoning, rhetoric, and role-induced deference intact, so its effect is empirical and conditional. Section 7 finds it real but confined to cooperative `deepseek-v4-flash`, with a counterfactual-answer probe ruling out the explanation that agents simply copy the visible answer.

## 5.2 Detect: Deliberation-Speed Trust

When overconfidence cannot be prevented, it can still be detected at deployment without labels. Corollary 1 shows that terminal confidence loses rank information under collapse, while the pre-collapse trajectory can still carry signal. The most informative signal is consensus speed: a consensus reached slowly, after the panel is argued into it, reflects assimilation under pressure, whereas one reached immediately reflects genuine agreement (verified in Section 7).

**Definition 3** (Deliberation-Speed Trust (Deliberation)). From a debate transcript, without the test label, compute label-free trajectory features: the self-reported confidence and answer-consensus at the first and last turns, the number of turns the modal answer changes, the change in confidence, and the *time to consensus*, the first turn at which agreement crosses a threshold. The *trust score* $\tau(x) \in [0,1]$ is a logistic model of these features, fit on a labeled calibration set of other conditions to predict answer correctness. At deployment $\tau$ is computed from the transcript alone and used for selective prediction: commit when $\tau$ is high, abstain or escalate when it is low.

Deliberation is a logistic head on seven label-free trajectory features. Its lift over self-reported confidence comes from the consensus-dynamics block, not any single feature. An ablation in Section 7 shows that dropping any one feature moves AUROC by $< 0.004$, echoing Corollary 1. It is trained on labels, but for other conditions, so it transfers to unseen domains and to `qwen-flash` with no labels for the deployment condition. It helps where confidence has gone rank-uninformative and adds nothing on `GLM-5.1`, whose confidence still ranks correctness. Whether to deploy it is itself decidable without labels: Corollary 1 points to the loss of terminal-confidence dynamic range, so a gate on the label-free dispersion $\hat{D} = \mathrm{SD}_x[C_T(x)]$ deploys Detect when $\hat{D}$ is small and confidence has collapsed flat, and keeps self-reported confidence otherwise. On the five evaluated settings this gate matches the better of the two, with the threshold selected on source settings and checked by leave-one-model-out validation (Section 7, Table 10).

## 5.3 Certify: Conformal Verdict Affirmation

Certify wraps the committed answer in a distribution-free coverage guarantee under exchangeability. It requires a labeled calibration split from the deployment distribution. Affirm certifies that the true label lies in the returned set with probability $\geq 1-\alpha$, not that any single answer is correct. Because it reads no internal signal, collapse cannot invalidate its coverage when calibration and test instances remain exchangeable.

**Definition 4** (Debate nonconformity). Fix a turn $t$. Let $\mathcal{V}_t$ be the pool of votes used for certification. This pool contains the $N$ agents' votes, or all agent–replicate votes when replicate runs are pooled. Define $\hat{p}_k(t) = |\{v \in \mathcal{V}_t : v = k\}|/|\mathcal{V}_t|$, and define the nonconformity of class $k$ as $s_k(t) = 1 - \hat{p}_k(t)$. Given a calibration set of $n$ labeled instances with true-label scores $\{s_{y^{(j)}}^{(j)}(t)\}_{j=1}^n$, let $k_\alpha = \lceil (n+1)(1-\alpha) \rceil$ and let $\hat{q}_\alpha(t)$ be the $k_\alpha$-th smallest calibration score (with $\hat{q}_\alpha(t) = +\infty$ if $k_\alpha > n$). The *conformal set* is $\widehat{\mathcal{C}}_\alpha(t) = \{k : s_k(t) \leq \hat{q}_\alpha(t)\}$.

**Theorem 4** (Marginal coverage under collapse). *If the calibration instances and the test instance are exchangeable, then for* any *fixed turn $t$, $\mathbb{P}(y \in \widehat{\mathcal{C}}_\alpha(t)) \geq 1 - \alpha$, irrespective of role design, task, or the value of $\rho(t)$. The guarantee uses only exchangeability of the nonconformity scores, never the (corrupted) agreement level.*

Theorem 4 is split conformal (Vovk et al., 2005); coverage holds even as $\rho(t) \to 1$ and $\widetilde{C}(t) \to 1$, so under exchangeability the conformal set is the one signal that survives diversity collapse. Proof in Appendix B.7.

## 5.4 Certified stopping and the safety separation

Affirm (Algorithm 1, Appendix C) stops at a turn $t^\star$ chosen on a selection fold, with the deployment threshold formed on an independent fold. Since $t^\star$ depends on neither the threshold fold nor the test point, the data-dependent stop keeps marginal coverage.

**Theorem 5** (Coverage at the calibration-selected stop). *Suppose $t^\star$ is chosen using only the selection fold $\mathcal{D}_{\mathrm{sel}}$, and the threshold $\hat{q}(t^\star)$ is computed on an independent fold $\mathcal{D}_{\mathrm{thr}}$ that is exchangeable with the test*

*instance. Then $t^\star$ is independent of $\mathcal{D}_{\text{thr}}$ and of the test point, so the fixed-turn guarantee (Theorem 4) applies directly at the now-fixed index $t^\star$:*

$$\mathbb{P}\big(y \in \widehat{\mathcal{C}}_\alpha(t^\star)\big) \;\geq\; 1 - \alpha, \tag{5}$$

*with no Bonferroni penalty and no dependence on the agreement level or the diversity collapse.*

Proof in Appendix B.8. Because the certificate reads no internal signal, augmenting the debate with any Prevent or Detect module changes only efficiency, not the coverage guarantee under the stated exchangeability conditions (Proposition 3, Appendix C).

Against the standard agreement heuristic, the separation is provable.

**Corollary 3** (Safety separation from agreement-stopping). *Let agreement-stopping return the modal answer once consensus $\kappa(t) \geq \theta$, and write $\tau_\theta$ for that stopping turn. Its singleton miscoverage is $R_{\tau_\theta} = 1 - A(\tau_\theta)$. In the collapse-limit version, or whenever the committed modal answer has already stabilized, $R_{\tau_\theta} = R_\infty$. Affirm's returned set has miscoverage $\leq \alpha$ at its calibration-selected stopping turn (Theorem 5). Hence*

$$\text{err}(agreement\text{-}stop) - \text{err}(Affirm) \;\geq\; R_{\tau_\theta} - \alpha, \tag{6}$$

*which is positive whenever the stopping-time residual exceeds $\alpha$; in the collapse-limit case this is $R_\infty - \alpha$. On hard or ambiguous tasks where this residual is large, agreement-stopping is much worse than its nominal confidence suggests, while Affirm stays at its guarantee. Equivalently, no agreement-only commit rule with a fixed consensus threshold $\theta < 1$ attains miscoverage $\leq \alpha$ uniformly over the family of collapsed tasks whose committed residual exceeds $\alpha$: no agreement threshold can be made uniformly safe there, so a certificate, not a better threshold, is required.*

Agreement-stopping trusts exactly the quantity the theory shows is inflated $(\kappa, \widetilde{C})$, and in the collapse-limit case the gap between the two procedures is the residual error $R_\infty$, the same quantity that equals the manufactured overconfidence (Theorem 2). Proof in Appendix B.9.

## 6 Experimental Setup

Our primary model is `deepseek-v4-flash` (DeepSeek-AI, 2026). We run the non-thinking setting with $N{=}3$ agents, $T{=}8$ turns, and $B{=}3$ replicates. The core benchmark is a controlled $15{\times}2$ grid: fifteen multiple-choice tasks, thirteen difficulty-spanning Massive Multitask Language Understanding (MMLU) domains (Hendrycks et al., 2021), TruthfulQA (Lin et al., 2022), and the AI2 Reasoning Challenge (ARC-Challenge) (Clark et al., 2018), each run under *cooperative* critique roles and *adversarial* persuasion roles on the *same* questions, giving thirty conditions. To check the findings are not single-model artifacts we replicate on four further families (`qwen-flash` (Qwen Team, Alibaba, 2026a), `GLM-5.1` (Zhipu AI, 2026), `Llama-3.3-70B` (Dubey et al., 2024), `GPT-4o-mini` (OpenAI, 2024)) and one reasoning-tuned model (`QwQ-plus` (Qwen Team, Alibaba, 2026b)), and to probe generality beyond multiple choice we add ReClor (Yu et al., 2020), CommonsenseQA (Talmor et al., 2019), and the free-form benchmarks GSM8K (Cobbe et al., 2021) and TriviaQA (Joshi et al., 2017). Exact API model strings, providers, decoding parameters, the confidence-elicitation format, and the answer-parsing rules are in Appendix D. Each agent logs its answer and confidence *separately*, so answer agreement is never computed from correctness. We report accuracy $A_T$, mean confidence $C_T$, the overconfidence gap $G_T = C_T - A_T$ and residual $R = 1 - A_T$, the calibration change $\Delta$ECE, the mechanism signals $\kappa(t)$ and $\rho(t)$, and, for the deployment components, selective-prediction AUROC, conformal miscoverage, and conformal set size. The task list, role designs, per-domain $Q$, and all metric definitions are in Appendix D.

## 7 Results

We test the diagnostic claim on the thirty-condition grid in Table 6. Consensus reaches $\bar{\kappa}(T) = 0.997$ and confidence compresses into $[0.90, 1.0]$ with mean 0.96, while accuracy spans 0.57–0.99 (Figure 3, left). The measured mechanism matches Lemma 1: $\bar{\rho}$ rises $0.53 \to 0.96$, score dispersion vanishes, and aggregate variance climbs toward the single-agent variance (Figure 2; error $1.4 \times 10^{-5}$). Thus the collapse is not just

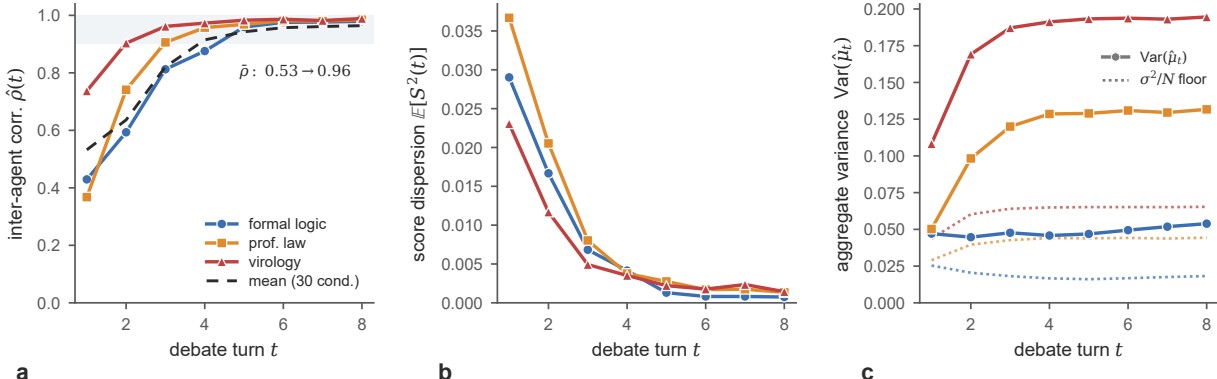

Figure 2: **Diversity collapse, measured directly.** Results are for `deepseek-v4-flash`; the dashed line is the mean over all thirty conditions. The panels show the inter-agent score correlation $\hat{\rho}(t)$, the score dispersion $\mathbb{E}[S^2(t)]$, and the aggregate variance $\mathrm{Var}(\hat{\mu}_t)$ against its $\sigma^2(t)/N$ floor over the debate horizon.

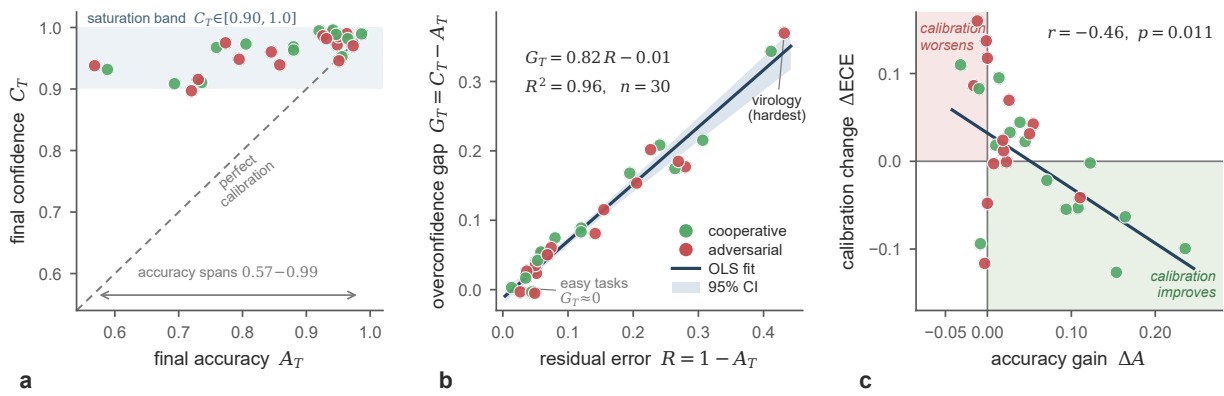

Figure 3: The three theory predictions across thirty conditions. Green points are cooperative roles and red points are adversarial roles. The panels show final confidence vs. accuracy (Theorem 1), the overconfidence gap vs. residual error (Theorem 2, Corollary 2), and calibration change vs. accuracy gain (Theorem 3).

final-answer unanimity; it is the covariance movement that removes the averaging benefit while suppressing visible disagreement.

Because $G_T = C_T - A_T$ and $R = 1 - A_T$ share $A_T$, the claim is not a raw correlation. Once confidence is nearly flat, a high gap–residual fit is partly mechanical: if $C_T$ were independent noise with the observed variance ratio $\mathrm{Var}(C_T) = \mathrm{Var}(A_T)/17$, the induced $R^2$ would already be about $17/18$. The evidence is therefore the collapse of the shortfall $1 - C_T$: $\mathrm{Var}(C_T)$ is $17\times$ smaller than $\mathrm{Var}(A_T)$, and one shared $\bar{\varepsilon} = 0.037$ predicts $\hat{G} = R - \bar{\varepsilon}$ out of sample better than fitted-slope transfer baselines by root-mean-square error: 0.010 vs. 0.022 on `qwen-flash`; 0.024 vs. 0.034 on `GLM-5.1`. As a diagnostic consequence, the primary grid has slope 0.82, CI $[0.73, 0.88]$, intercept $-0.01$, and $R^2 = 0.96$ across residuals 0.01–0.43 (Figure 3, center). The relation is already formed by turn 3 and remains stable through turn 8 (Table 13); calibration improves only when accuracy does ($r = -0.46$, $p = 0.011$). Lone-agent controls attribute about half of the inflation, and the consensus collapse itself, to debate-specific assimilation rather than baseline overconfidence (Appendix E).

CMAD separates certification from agreement. Agreement-based stopping commits the modal answer and therefore has miscoverage at the residual rate, 0.18–0.47 on hard domains. Affirm reads no agreement signal and uses an exchangeable labeled calibration split; under that condition it holds marginal set coverage, with means 0.964 on `deepseek-v4-flash` and 0.961 on `qwen-flash`. On the same transcripts (Table 1), every

Table 1: **Agreement-based and adaptive stopping is unsafe under collapse.** Results are for `deepseek-v4-flash` on 16 hard conditions; entries report mean$\pm$SD.

| Rule | Error & calibration | | | Efficiency | | |
|---|---|---|---|---|---|---|
| | miscov$\downarrow$ | singl-err$\downarrow$ | ECE$\downarrow$ | $|\mathcal{C}|\downarrow$ | turns$\downarrow$ | commit%$\uparrow$ |
| *A. Agreement / stability stopping* | | | | | | |
| fixed-$T$ | $0.22_{\pm0.09}$ | $0.22_{\pm0.09}$ | $0.17_{\pm0.09}$ | $1.00_{\pm0.00}$ | $8.0_{\pm0.0}$ | 100% |
| agreement@0.9 | $0.22_{\pm0.09}$ | $0.22_{\pm0.09}$ | $0.15_{\pm0.07}$ | $1.00_{\pm0.00}$ | $2.4_{\pm0.6}$ | 100% |
| confidence@0.9 | $0.22_{\pm0.09}$ | $0.22_{\pm0.09}$ | $0.15_{\pm0.09}$ | $1.00_{\pm0.00}$ | $3.2_{\pm1.1}$ | 100% |
| stability@2 | $0.22_{\pm0.09}$ | $0.22_{\pm0.09}$ | $0.15_{\pm0.08}$ | $1.00_{\pm0.00}$ | $2.1_{\pm0.1}$ | 100% |
| *B. Budget / trigger stopping* | | | | | | |
| single-model-trigger@0.9 | $0.22_{\pm0.09}$ | $0.22_{\pm0.09}$ | $0.16_{\pm0.09}$ | $1.00_{\pm0.00}$ | $6.5_{\pm1.4}$ | 100% |
| early-exit (no-change) | $0.22_{\pm0.09}$ | $0.22_{\pm0.09}$ | $0.15_{\pm0.08}$ | $1.00_{\pm0.00}$ | $2.1_{\pm0.1}$ | 100% |
| *C. Certified (reads no agreement)* | | | | | | |
| conformal@$T$ (Wang et al., 2026) | $0.03_{\pm0.05}$ | $0.12_{\pm0.01}$ | $0.10_{\pm0.00}$ | $3.45_{\pm1.15}$ | $8.0_{\pm0.0}$ | 18% |
| MICP budget (Zhou et al., 2026) | $0.02_{\pm0.04}$ | $0.12_{\pm0.01}$ | $0.10_{\pm0.05}$ | $3.64_{\pm0.96}$ | $7.2_{\pm2.2}$ | 12% |
| **Affirm** (ours) | $0.05_{\pm0.06}$ | $0.10_{\pm0.04}$ | $0.08_{\pm0.04}$ | $2.72_{\pm1.33}$ | $1.4_{\pm0.9}$ | 34% |

Table 2: **Deliberation-Speed Trust (*Detect*)** vs. self-reported confidence: AUROC, selective accuracy at 70% coverage, and condition-wise wins. Within-model rows use CV; the `deepseek` win count is leave-one-domain-out.

| Deployment target | votes/turn | AUROC$\uparrow$ | | Acc@70%$\uparrow$ | | Delib. wins$\uparrow$ |
|---|---|---|---|---|---|---|
| | | conf | Delib | conf | Delib | |
| `deepseek` (within, CV) | 9 | 0.756 | 0.793 | 0.943 | 0.952 | 14/19 (LODO) |
| `qwen-flash` (transfer) | 9 | 0.728 | 0.770 | 0.878 | 0.910 | 24/34 |
| `qwen-flash` (transfer) | 3 | 0.656 | 0.714 | 0.850 | 0.879 | 27/34 |
| `GLM-5.1` | 3 | 0.840 | 0.807 | 0.824 | 0.823 | 3/16 |

Table 3: **Cross-model generality.** Collapse, gap-vs-residual fit, Affirm coverage, and Sequester $\Delta$ECE where run. A dash denotes not evaluated.

| Model | $\bar{\rho}\,(1{\to}T)$ | Gap–residual relation | | | Affirm cov$\uparrow$ | Sequester $\Delta$ECE$\downarrow$ | |
|---|---|---|---|---|---|---|---|
| | | slope | $R^2\uparrow$ | const-RMSE$\downarrow$ | | coop$\downarrow$ | adv$\downarrow$ |
| `deepseek-v4-flash` | $0.53{\to}0.96$ | 0.82 | 0.96 | 0.021 | 0.964 | $-0.046$ | $-0.010$ |
| `qwen-flash` | $0.61{\to}0.97$ | 0.96 | 1.00 | 0.010 | 0.961 | $+0.010$ | $-0.011$ |
| `GLM-5.1` | $0.65{\to}0.94$ | 0.92 | 0.99 | 0.024 | 0.992 | | — |
| `Llama-3.3-70B` | $0.51{\to}0.97$ | 0.84 | 0.89 | 0.034 | 1.00 | | — |
| `GPT-4o-mini` | $0.55{\to}0.88$ | 0.76 | 0.76 | 0.061 | 1.00 | | — |
| `QwQ-plus` | $0.73{\to}0.93$ | 0.92 | 0.99 | 0.024 | 1.00 | | — |

agreement- or confidence-reading rule commits confident wrong answers at miscov $\approx 0.22$ and ECE 0.15–0.17; only the certified family holds $\leq \alpha$ on the exchangeable split. Affirm is also earliest on average (1.4 vs. 7–8 turns), but it commits to a singleton only 34% of the time and returns larger sets rather than unsafe singletons when it cannot certify. This is a coverage result, not a hidden point-prediction gain: the certified system is allowed to abstain by widening the set when agreement is untrustworthy.

The signal-based components are conditional. *Detect* uses consensus speed: correctness falls from 0.94 at immediate consensus to 0.55 after seven turns, raising AUROC on `deepseek-v4-flash` ($0.756 \to 0.793$) and transferred `qwen-flash` ($0.728 \to 0.770$), but not on `GLM-5.1`, whose confidence still ranks (Table 2). A label-free gate based on $\hat{D} = \text{SD}_x[C_T(x)]$ abstains on this boundary case. *Prevent* cuts cooperative `deepseek-v4-flash` ECE by 0.046 at no accuracy cost, but the gain vanishes on `qwen-flash`; trajectory examples are in Appendix E.7.

Table 4: **Heterogeneous vs. homogeneous debate.** Cooperative high-residual domains ($Q = 60, T = 8$). The mixed panel starts more diverse but collapses to the same terminal correlation, confidence, and gap.

| Panel | Kind | $A\uparrow$ | $C_T$ | $G_T\downarrow$ | $1 - A$ | $\bar{\rho}(1)$ | $\bar{\rho}(T)$ |
|---|---|---|---|---|---|---|---|
| deepseek-v4-flash×3 | homog. | 0.72 | 0.93 | 0.21 | 0.28 | 0.68 | 0.99 |
| qwen-flash×3 | homog. | 0.60 | 0.96 | 0.35 | 0.40 | 0.65 | 0.96 |
| glm-5.1×3 | homog. | 0.71 | 0.97 | 0.26 | 0.29 | 0.73 | 0.96 |
| DeepSeek+Qwen+GLM | hetero | 0.69 | 0.96 | 0.28 | 0.31 | **0.44** | **0.96** |

The mechanism also survives the natural fix of using different models. A heterogeneous `DeepSeek`/`Qwen`/`GLM` panel starts more independent than any homogeneous panel ($\bar{\rho}(1) = 0.44$ vs. 0.65–0.73), but debate drives it to the same terminal correlation, confidence, and gap (Table 4). The failure is therefore endogenous to the interaction, not merely to identical agents. The first-turn gap shows that model diversity is real; the terminal equality shows that the interaction removes that independence.

The distribution-level pattern reproduces in additional model-family probes (Table 3): confidence saturates, correlation collapses, and the gap tracks the residual with sub-unit slopes 0.96, 0.92, 0.84, and 0.76. Agreement-stopping remains unsafe, with hardest-case miscoverage above 0.50, while Affirm holds exchangeable-split coverage in the 0.961–1.00 range. `GPT-4o-mini` is the weakest boundary case: its larger shortfall $\bar{\varepsilon} = 0.085$ lowers the fit to $R^2 = 0.76$, but residual-tracking remains monotone. These rows are replication probes rather than equally powered estimates; the primary grid supplies the full thirty-condition replicated estimate. Appendices E and F extend the signature to `QwQ-plus`, new task types, free-form generation, exogenous-signal stress tests, and $N \in \{5, 7\}$.

## 8  Discussion and Limitations

LLM overconfidence is known (Guo et al., 2017; Tian et al., 2023); our contribution is the mechanism under debate. The empirical content is not the algebra $G = R - \varepsilon$, but the measured collapse of $\varepsilon = 1 - C_T$ to a near-constant: one shared shortfall predicts per-condition gaps out of sample. The theory assumes diversity collapse, which we measure directly, and proves collapse for dispersion-based *apparent* confidence; verbalized confidence is a separate quantity whose co-saturation we verify rather than assume. The study spans nineteen tasks, five model families, free-form short-answer settings, and $N \in \{3, 5, 7\}$; long-form generation, larger scales, and model-internal confidence channels remain untested. A retrieval- and tool-augmented mechanism test shows that the effect is not merely text-only: when an exogenous signal keeps agents independent, collapse is arrested and the gap stops tracking the residual (Appendix F). Prevent and Detect remain conditional, although the Detect gate routes every model correctly under leave-one-model-out validation. Finally, $R_\infty$ is only the residual left by this finite-horizon debate, and conformal coverage requires exchangeability; cross-domain calibration is therefore a stress test, not a deployment guarantee (Appendix F).

Debate is still worth running when roles critique and the task has headroom: in those regimes it raises accuracy and improves calibration. The practical lesson is to separate answer generation from answer certification. CMAD does so by preventing coupling where verdict exposure matters, detecting residual coupling from trajectories when confidence has lost rank information, and certifying answers with split conformal prediction when coverage is required. The signal-based levers are conditional; under exchangeability, the certificate is the collapse-unconditional backstop. More broadly, self-consistency decoding, majority voting over sampled reasoning, and LLM-as-judge pipelines all risk treating concordance as independent corroboration after a shared or mutually-conditioning process has made it endogenous. Even a three-model panel starts more independent but collapses to the same terminal gap (Table 4). The question is therefore not only whether a system agrees, but whether that agreement was formed independently of the answer it is trusted to certify. Agreement can propose an answer. It should not certify one it helped manufacture.

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

## Appendix Contents

# A    Notation

Table 5: **Notation.** Symbols used in the paper.

| Symbol | Meaning |
|---|---|
| *Roman symbols* | |
| $A(t), A_t$ | Accuracy of the aggregated (modal) answer at turn $t$; $A_\infty$ its limit, $A_T$ the terminal value. |
| AUROC | Area under the ROC curve for correctness ranking. |
| $B$ | Number of replicates per condition in the experiments. |
| $c_i(t)$ | Agent $i$'s self-reported confidence at turn $t$, in $[0,1]$. |
| $C(t), C_t$ | Reported (panel) confidence, $\frac{1}{N}\sum_i c_i(t)$; $C_\infty = 1 - \varepsilon$ its limit. |
| $\widetilde{C}(t)$ | Apparent (agreement-based) confidence an operator reads from observed agreement. |
| $\widehat{\mathcal{C}}_\alpha(t)$ | Conformal prediction set at turn $t$ and level $\alpha$; $|\widehat{\mathcal{C}}_\alpha|$ its size. |
| $\mathcal{D}_{\mathrm{sel}}, \mathcal{D}_{\mathrm{thr}}$ | Selection fold (for the stopping turn) and threshold fold (for the conformal quantile). |
| ECE | Expected calibration error over 10 equal-width bins; $\Delta\mathrm{ECE} = \mathrm{ECE}_T - \mathrm{ECE}_1$. |
| $G(t), G_t$ | Calibration / overconfidence gap, $C(t) - A(t)$; $G_\infty$ the terminal gap. |
| $K$ | Number of answer options (classes). |
| $\ell(t)$ | Loss, $1 - A(t)$. |
| $N$ | Number of agents in the panel. |
| $\hat{p}_k(t)$ | Pooled vote fraction for class $k$ over the certification vote pool $\mathcal{V}_t$. |
| $\mathcal{P}$ | Instance distribution (questions drawn i.i.d.). |
| $\mathcal{V}_t$ | Vote pool used by Certify at turn $t$ (agents, or agent–replicate votes if pooled). |
| $\hat{q}_\alpha(t)$ | Split-conformal order-statistic threshold at level $\alpha$. |
| $Q$ | Number of questions per condition. |
| $R_t, R, R_\infty$ | Current error $R_t = 1 - A(t)$; terminal residual proxy $R = 1 - A_T$; collapse-limit residual $R_\infty = \lim_t \ell(t)$. |
| $s_k(t)$ | Nonconformity score of class $k$, $1 - \hat{p}_k(t)$. |
| $S^2(t)$ | Score dispersion across agents, $\frac{1}{N}\sum_i \big(X_i(t) - \hat{\mu}_t\big)^2$. |
| $t, T$ | Turn index and horizon (number of turns); $t^\star$ the calibration-selected stopping turn. |
| $X_i(t)$ | Per-agent correctness score, $\phi(\hat{y}_i(t), c_i(t)) \in [0,1]$. |
| $y, \mathcal{Y}$ | Gold (true) label and the answer space $\{1,\dots,K\}$. |
| $\hat{y}_i(t)$ | Agent $i$'s answer at turn $t$; $\hat{y}^{\mathrm{agg}}(t)$ the aggregated modal answer. |
| *Greek symbols* | |
| $\alpha$ | Conformal miscoverage level, for example 0.1. |
| $\Delta A, \Delta C, \Delta G$ | Change of a quantity over the horizon, $(\cdot)(T) - (\cdot)(1)$. |
| $\varepsilon$ | Confidence shortfall, $1 - C_\infty$; $\bar{\varepsilon}$ the single shared constant ($\approx 0.037$). |
| $\kappa(t)$ | Answer consensus: fraction of votes on the modal answer. |
| $\hat{\mu}_t$ | Aggregate score, $\frac{1}{N}\sum_i X_i(t)$. |
| $\rho(t)$ | Inter-agent (equi)correlation of $X_i(t)$; $\rho_\infty$ its limit (collapse if $\rho_\infty = 1$). |
| $\sigma^2(t)$ | Per-agent variance of $X_i(t)$; $\sigma^2_{\max}$ its bound. |
| $\tau(x)$ | Deliberation trust score in $[0,1]$ (label-free trajectory features). |
| $\phi(\cdot)$ | Correctness-score map ($c$ if the answer is correct, $1 - c$ otherwise). |
| *Acronyms* | |
| CMAD | Calibrated Multi-Agent Debate (the framework). |
| Certify / Affirm | Conformal Verdict Affirmation: the exogenous coverage guarantee. |
| Detect / Deliberation | Deliberation-Speed Trust: label-free trajectory-based trust score. |
| ICC | Intraclass correlation (one-way random-effects estimator for $\rho$). |
| Prevent / Sequester | Verdict-Sequestered Debate: peers' committed verdicts withheld. |

# B  Proofs

## B.1  The collapse-dynamics identity (Lemma 1)

*Proof.* Fix $t$ and drop it from the notation. With $\mathrm{Var}(X_i) = \sigma^2$ and $\mathrm{Cov}(X_i, X_j) = \rho\sigma^2$ for $i \neq j$,

$$\mathrm{Var}(\hat{\mu}) = \frac{1}{N^2}\left[\sum_i \mathrm{Var}(X_i) + \sum_{i\neq j}\mathrm{Cov}(X_i, X_j)\right] = \frac{1}{N^2}\left[N\sigma^2 + N(N-1)\rho\sigma^2\right] = \frac{\sigma^2}{N}\left[1 + (N-1)\rho\right]. \quad \text{(B.1.1)}$$

For the dispersion, $\sum_i (X_i - \hat{\mu})^2 = \sum_i X_i^2 - N\hat{\mu}^2$, so by linearity

$$\mathbb{E}\left[\sum_i (X_i-\hat{\mu})^2\right] = \sum_i \mathbb{E}[X_i^2] - N\,\mathbb{E}[\hat{\mu}^2] = N(\sigma^2+\mu^2) - N\left(\frac{\sigma^2}{N}[1+(N-1)\rho]+\mu^2\right) = \sigma^2(N-1)(1-\rho), \quad \text{(B.1.2)}$$

using $\mathbb{E}[X_i^2] = \sigma^2 + \mu^2$ and $\mathbb{E}[\hat{\mu}^2] = \mathrm{Var}(\hat{\mu}) + \mu^2$. Dividing by $N$ gives $\mathbb{E}[S^2] = \frac{N-1}{N}\sigma^2(1-\rho)$. $\qquad\square$

## B.2  Confidence saturation (Theorem 1)

*Proof.* By Lemma 1,

$$\mathbb{E}[S^2(t)] = \frac{N-1}{N}\,\sigma^2(t)\,(1-\rho(t)). \quad \text{(B.2.1)}$$

Since $\sigma^2(t) \leq \sigma_{\max}^2$ and $\rho(t) \to 1$, the right-hand side $\to 0$, so $S^2(t) \to 0$ in $L^1$ and hence in probability. The observed disagreement also vanishes in this limit by Assumption 1, and the apparent confidence $\widetilde{C}(t)$ is a continuous statistic of that disagreement (Definition 1), maximal at unanimity, so the continuous mapping theorem gives $\widetilde{C}(t) \to 1$. No step uses the residual error $R_\infty$ or the bias, so the limit is independent of correctness. $\qquad\square$

## B.3  Terminal agreement loses its rank signal (Corollary 1)

*Proof.* Write $C_\infty(x)$ for the terminal confidence on instance $x$ of a fixed task. For apparent confidence this is Theorem 1; for reported confidence assume the empirical co-saturation condition stated in Corollary 1. Then $C_\infty(x)$ concentrates at a common ceiling for every $x$, so $\mathrm{Var}_x[C_\infty(x)] \to 0$: the score concentrates at that ceiling. Recall the rank statistic

$$\mathrm{AUROC} = \mathbb{P}\big(C_\infty(x^+) > C_\infty(x^-)\big) + \tfrac{1}{2}\mathbb{P}\big(C_\infty(x^+) = C_\infty(x^-)\big), \quad \text{(B.3.1)}$$

where $x^+, x^-$ are an independent correct and incorrect instance. At exact saturation ($C_\infty$ constant in $x$) every such pair is a tie, so $\mathrm{AUROC} = \frac{1}{2}$ exactly. Off this degenerate point the only possible ranking information is the shortfall $\varepsilon(x) = 1 - C_\infty(x)$, since $C_\infty(x^+) > C_\infty(x^-)$ iff $\varepsilon(x^+) < \varepsilon(x^-)$. AUROC is therefore *not* a function of $\mathrm{Var}_x[C_\infty]$: a vanishing $\varepsilon$ can still rank the labels perfectly, for example if $\varepsilon(x) = 0$ on correct instances and $\delta$ on incorrect instances for every $\delta > 0$. Variance collapse therefore does not by itself force the AUROC to $\frac{1}{2}$; it guarantees only that whatever discrimination remains lives entirely in the vanishing $\varepsilon(x)$, which carries reliable rank information only if its sign relative to correctness is systematic. Empirically it is near chance on the hard tasks (within-task AUROC 0.49–0.58), so terminal confidence is at best a weak ranker there. In contrast a pre-collapse statistic such as the dispersion $S^2(t)$ at small $t$, or the turns-to-consensus, has $\mathrm{Var}_x > 0$ because $\rho(t) < 1$ there, so it can retain rank information; this is what the *Detect* component exploits. $\qquad\square$

## B.4  Overconfidence tracks the residual (Theorem 2 and Corollary 2)

*Proof.* By definition $G_\infty = C_\infty - A_\infty$ and $A_\infty = 1 - R_\infty$, so

$$G_\infty = C_\infty - 1 + R_\infty = R_\infty - (1 - C_\infty). \quad \text{(B.4.1)}$$

When reported confidence co-saturates with the apparent signal, $C_\infty \to 1$ in the ideal limit, whence $G_\infty \to R_\infty$. Writing the confidence shortfall $\varepsilon = 1 - C_\infty \geq 0$, we have

$$G_\infty = R_\infty - \varepsilon. \quad \text{(B.4.2)}$$

If $\varepsilon$ is constant across tasks the relation is affine in $R_\infty$ with unit slope and intercept $-\varepsilon$. More generally, for two tasks with $R_1 < R_2$ and shortfalls $\varepsilon_1, \varepsilon_2$,

$$G_\infty(R_2) - G_\infty(R_1) = (R_2 - R_1) - (\varepsilon_2 - \varepsilon_1). \tag{B.4.3}$$

Thus $G_\infty$ is strictly increasing whenever $0 \le \varepsilon_2 - \varepsilon_1 < R_2 - R_1$, which is Corollary 2. If $\varepsilon$ is differentiable in $R_\infty$, this finite-difference condition becomes $0 \le \mathrm{d}\varepsilon/\mathrm{d}R_\infty < 1$ and the local slope is $1 - \mathrm{d}\varepsilon/\mathrm{d}R_\infty \in (0, 1]$. The empirical slope $0.82 \in (0, 1]$ is consistent with a small positive shortfall slope. $\square$

### B.5 Finite-time overconfidence rate (Proposition 1)

**Proposition 1** (Finite-time overconfidence rate). *Let the apparent confidence be a dispersion map $\widetilde{C}(t) = g(\mathbb{E}[S^2(t)])$ with $g(0) = 1$, $g$ nonincreasing and $L$-Lipschitz on $[0, \sigma_{\max}^2]$, write $R_t = 1 - A(t)$ for the current error, and let $\widetilde{G}(t) = \widetilde{C}(t) - A(t)$ be the apparent gap. Then at every turn,*

$$R_t - L \, \tfrac{N-1}{N} \, \sigma^2(t) \, (1 - \rho(t)) \ \le \ \widetilde{G}(t) \ \le \ R_t. \tag{B.5.1}$$

*Hence the apparent overconfidence is within $L \, \mathbb{E}[S^2(t)]$ of the current error already at turn $t$, contracting at the rate of the measurable collapse factor $1 - \rho(t)$ and recovering $\widetilde{G}_\infty = R_\infty$ as $\rho(t) \to 1$ (Theorem 2).*

The bound governs dispersion-based confidences ($\widetilde{C} = 1 - S^2$ gives $L = 1$, where the lower bound is tight); the answer-consensus $\kappa$ and the reported confidence $C$ need not be Lipschitz functions of $S^2$ and are not covered.

*Proof.* By $g(0) = 1$, monotonicity, and the Lipschitz bound, $1 - \widetilde{C}(t) = g(0) - g(\mathbb{E}[S^2(t)]) \in [0, L\,\mathbb{E}[S^2(t)]]$; with $\widetilde{G}(t) = \widetilde{C}(t) - A(t) = R_t - (1 - \widetilde{C}(t))$ this gives the two bounds after substituting $\mathbb{E}[S^2(t)] = \tfrac{N-1}{N}\sigma^2(t)(1 - \rho(t))$ (Lemma 1). $\square$

### B.6 Productive–overconfident dichotomy (Theorem 3)

*Proof.* By definition $G(t) = C(t) - A(t)$, hence $\Delta G = G(T) - G(1) = \Delta C - \Delta A$. The stated confidence-inflation condition gives $\Delta C = C(T) - C(1) \ge 0$. Therefore

$$\Delta G < 0 \iff \Delta A > \Delta C \ge 0. \tag{B.6.1}$$

Because $\Delta C \ge 0$, a necessary condition for decreasing the overconfidence gap ($\Delta G < 0$) is $\Delta A > 0$: debate that does not raise accuracy cannot decrease the gap, and strictly increases it ($\Delta G = \Delta C > 0$) whenever the collapse inflates confidence ($\Delta C > 0$) with no accuracy gain ($\Delta A = 0$). $\square$

### B.7 Split-conformal coverage (Theorem 4)

*Proof.* Fix a turn $t$ and set $k_\alpha = \lceil (n+1)(1-\alpha) \rceil$. The calibration scores $\{s_{y^{(j)}}^{(j)}(t)\}_{j=1}^n$ and the test true-label score $s_y^{\text{test}}(t)$ are obtained by applying the same turn-$t$ procedure to the calibration and test instances; under exchangeability of the instances they are exchangeable. If $k_\alpha > n$ then $\hat{q}_\alpha(t) = +\infty$ and coverage is automatic, so assume $k_\alpha \le n$. With arbitrary randomized tie-breaking, the rank of $s_y^{\text{test}}(t)$ among the $n + 1$ scores is uniform on $\{1, \ldots, n+1\}$. If this rank is at most $k_\alpha$, then $s_y^{\text{test}}(t)$ is no larger than the $k_\alpha$-th smallest calibration score $\hat{q}_\alpha(t)$; therefore

$$\mathbb{P}\big(s_y^{\text{test}}(t) \le \hat{q}_\alpha(t)\big) \ \ge \ \frac{k_\alpha}{n+1} \ \ge \ 1 - \alpha, \tag{B.7.1}$$

which is exactly the event $y \in \widehat{\mathcal{C}}_\alpha(t)$. No quantity in the argument depends on $\rho(t)$ or on the apparent confidence $\widetilde{C}(t)$, so the guarantee is unaffected by diversity collapse. $\square$

## B.8 Coverage at the selected stopping turn (Theorem 5)

*Proof.* The stopping turn $t^\star$ is a function of the selection fold $\mathcal{D}_{\text{sel}}$ only, hence independent of both the threshold fold $\mathcal{D}_{\text{thr}}$ and the test instance. Condition on $\mathcal{D}_{\text{sel}}$; then $t^\star = \tau$ is a fixed index, while $\mathcal{D}_{\text{thr}}$ and the test instance are unaffected by the conditioning (they are independent of $\mathcal{D}_{\text{sel}}$) and remain exchangeable. Applying Theorem 4 at the fixed turn $\tau$ with calibration fold $\mathcal{D}_{\text{thr}}$ gives $\mathbb{P}(y \in \widehat{\mathcal{C}}_\alpha(\tau) \mid \mathcal{D}_{\text{sel}}) \geq 1 - \alpha$; since this holds for every realisation of $\mathcal{D}_{\text{sel}}$, taking the expectation over $\mathcal{D}_{\text{sel}}$ preserves it,

$$\mathbb{P}(y \in \widehat{\mathcal{C}}_\alpha(t^\star)) \geq 1 - \alpha. \tag{B.8.1}$$

*Per-instance variant (Remark 1).* If $t^\star$ may depend on the test point, fix the level at $\alpha' = \alpha/T_{\max}$. For each fixed $t$, $\mathbb{P}(y \notin \widehat{\mathcal{C}}_{\alpha'}(t)) \leq \alpha'$, so a union bound gives $\mathbb{P}(\exists t : y \notin \widehat{\mathcal{C}}_{\alpha'}(t)) \leq T_{\max}\alpha' = \alpha$; thus the simultaneous event $E = \{y \in \widehat{\mathcal{C}}_{\alpha'}(t) \ \forall t\}$ has $\mathbb{P}(E) \geq 1 - \alpha$, and on $E$ coverage holds at the realised $t^\star$ for any selection rule. The anytime-valid form replaces the union bound by the optional-stopping property of a conformal test martingale (Ramdas et al., 2023). $\square$

## B.9 Safety separation from agreement-stopping (Corollary 3)

*Proof.* Let $\tau_\theta = \inf\{t : \kappa(t) \geq \theta\}$ be the agreement-stopping time. Under Assumption 1, $\kappa(t) \to 1$, so $\tau_\theta$ is finite for every $\theta < 1$ in the collapse limit. The procedure returns the singleton modal answer $\hat{y}^{\text{agg}}(\tau_\theta)$, whose miscoverage is

$$R_{\tau_\theta} = 1 - A(\tau_\theta) = \mathbb{P}(\hat{y}^{\text{agg}}(\tau_\theta) \neq y). \tag{B.9.1}$$

Affirm returns $\widehat{\mathcal{C}}_\alpha(t^\star)$ at the calibration-selected stopping turn $t^\star$ with $\mathbb{P}(y \notin \widehat{\mathcal{C}}_\alpha(t^\star)) \leq \alpha$ by the coverage guarantee (Theorem 5). Subtracting the two miscoverage probabilities,

$$\text{err(agreement-stop)} - \text{err(Affirm)} \ \geq \ R_{\tau_\theta} - \alpha. \tag{B.9.2}$$

If the modal answer has stabilized by $\tau_\theta$, or if one evaluates the collapse-limit commit rule, then $R_{\tau_\theta} = R_\infty$ and the displayed bound becomes $R_\infty - \alpha$. Thus the difference is positive whenever the residual at the committed turn exceeds $\alpha$. $\square$

## B.10 Collapse under general covariance (Proposition 2)

Equicorrelation (Lemma 1) is used only for closed forms; the collapse mechanism is structure-free. Let $X(t) \in \mathbb{R}^N$ have mean $\mathbb{E}[X(t)] = \mu(t)\mathbf{1}$ and covariance $\Sigma(t) \succ 0$ with bounded diagonal $\Sigma_{ii}(t) \leq \sigma_{\max}^2$. The debate aggregate is the simple mean $\hat{\mu}_t = \frac{1}{N}\mathbf{1}^\top X(t)$ (agents are weighted equally); under inverse-variance (GLS) weighting the weights are $w^\star = \Sigma(t)^{-1}\mathbf{1}/(\mathbf{1}^\top\Sigma(t)^{-1}\mathbf{1})$, which reduce to $\mathbf{1}/N$ under equicorrelation, so the two coincide in the symmetric case.

**Proposition 2** (Collapse under general covariance). *Suppose the covariance converges to rank one, $\Sigma(t) \to \sigma_\infty^2 \mathbf{1}\mathbf{1}^\top$ (all pairwise correlations $\to 1$ and all variances $\to \sigma_\infty^2$). Then:*

(i) *the aggregate variance loses the averaging benefit,* $\text{Var}(\hat{\mu}_t) = \dfrac{\mathbf{1}^\top\Sigma(t)\mathbf{1}}{N^2} \to \sigma_\infty^2$;

(ii) *every centered quadratic dispersion statistic $D(t) = X(t)^\top M X(t)$ with $M = M^\top \succeq 0$ and $M\mathbf{1} = \mathbf{0}$ vanishes in expectation,* $\mathbb{E}[D(t)] = \text{tr}(M\Sigma(t)) \to 0$.

*Consequently any dispersion-based apparent confidence satisfies $\widetilde{C}(t) \to 1$, and the apparent terminal gap satisfies $\widetilde{G}_\infty = R_\infty$. The reported gap remains $G_\infty = R_\infty - (1 - C_\infty)$ and tracks the residual only under the co-saturation condition of Theorem 2.*

*Proof. (i)* Writing $\hat{\mu}_t = \frac{1}{N}\mathbf{1}^\top X(t)$,

$$\text{Var}(\hat{\mu}_t) = \frac{1}{N^2}\mathbf{1}^\top\Sigma(t)\mathbf{1} \to \frac{1}{N^2}\sigma_\infty^2\mathbf{1}^\top\mathbf{1}\mathbf{1}^\top\mathbf{1} = \frac{\sigma_\infty^2}{N^2}(\mathbf{1}^\top\mathbf{1})^2 = \frac{\sigma_\infty^2}{N^2}N^2 = \sigma_\infty^2, \tag{B.10.1}$$

---

**Algorithm 1 Conformal Verdict Affirmation.** The Affirm stopping algorithm.

---

1: **Input:** max turns $T_{\max}$, level $\alpha$, calibration set split into a selection fold $\mathcal{D}_{\mathrm{sel}}$ ($|\mathcal{D}_{\mathrm{sel}}| = n_{\mathrm{sel}}$) and an independent threshold fold $\mathcal{D}_{\mathrm{thr}}$ ($|\mathcal{D}_{\mathrm{thr}}| = n$), tolerance $\epsilon \geq 0$.
2: **// Calibration phase (offline, label-aware).**
3: **for** $t = 1, \ldots, T_{\max}$ **do**
4:     $k_{\mathrm{sel}} \leftarrow \lceil (n_{\mathrm{sel}} + 1)(1 - \alpha) \rceil$; $\hat{q}_{\mathrm{sel}}(t) \leftarrow$ the $k_{\mathrm{sel}}$-th smallest true-label score $\{1 - \hat{p}_{y^{(j)}}(t)\}$ over $\mathcal{D}_{\mathrm{sel}}$ (or $+\infty$ if $k_{\mathrm{sel}} > n_{\mathrm{sel}}$)
5:     $\bar{s}(t) \leftarrow$ mean of $\big|\{k : 1 - \hat{p}_k(t) \leq \hat{q}_{\mathrm{sel}}(t)\}\big|$ over $\mathcal{D}_{\mathrm{sel}}$        {mean set size, selection fold only}
6: **end for**
7: $t^\star \leftarrow$ earliest $t$ with $\bar{s}(t) \leq \min_\tau \bar{s}(\tau) + \epsilon$            {$t^\star$ is a function of $\mathcal{D}_{\mathrm{sel}}$ only}
8: $k_{\mathrm{thr}} \leftarrow \lceil (n+1)(1-\alpha) \rceil$; $\hat{q}(t^\star) \leftarrow$ the $k_{\mathrm{thr}}$-th smallest score in $\{1 - \hat{p}_{y^{(j)}}(t^\star)\}$ over the independent fold $\mathcal{D}_{\mathrm{thr}}$ (or $+\infty$ if $k_{\mathrm{thr}} > n$)        {deployment threshold, held-out fold}
9: **// Deployment phase (per test instance).**
10: run debate to turn $t^\star$; **return** $\widehat{\mathcal{C}}_\alpha(t^\star) = \{k : 1 - \hat{p}_k(t^\star) \leq \hat{q}(t^\star)\}$; commit to the singleton iff $|\widehat{\mathcal{C}}_\alpha(t^\star)| = 1$.

---

using $\mathbf{1}^\top \mathbf{1} = N$. *(ii)* Since $X^\top M X$ is quadratic and $M\mathbf{1} = \mathbf{0}$,

$$\mathbb{E}[X^\top M X] = \mathrm{tr}(M\Sigma) + \mu^2 \, \mathbf{1}^\top M \mathbf{1} = \mathrm{tr}(M\Sigma), \tag{B.10.2}$$

and as $\Sigma \to \sigma_\infty^2 \mathbf{1}\mathbf{1}^\top$,

$$\mathrm{tr}(M\Sigma) \to \sigma_\infty^2 \, \mathrm{tr}(M\mathbf{1}\mathbf{1}^\top) = \sigma_\infty^2 \, \mathbf{1}^\top M \mathbf{1} = 0. \tag{B.10.3}$$

The centered correctness-score dispersion $S^2(t) = \frac{1}{N} \sum_i (X_i - \hat{\mu}_t)^2$ of Definition 1 is of this form (with $M = \frac{1}{N}(I - \frac{1}{N}\mathbf{1}\mathbf{1}^\top)$, which satisfies $M\mathbf{1} = \mathbf{0}$), so $\mathbb{E}[S^2(t)] \to 0$. The apparent-confidence saturation follows as in Theorem 1; the apparent gap is then $\widetilde{C}_\infty - A_\infty = 1 - (1 - R_\infty) = R_\infty$. For reported confidence the algebraic identity in Theorem 2 is unchanged, and residual-tracking again requires the stated co-saturation condition. $\qquad\square$

## C  The Affirm stopping algorithm and conformal variants

**Proposition 3** (CMAD safety invariant: efficiency–coverage decoupling). *Let the returned object be the conformal set $\widehat{\mathcal{C}}_\alpha(t^\star)$ of Algorithm 1. Augment the debate with any Prevent module (a transcript transformation such as Sequester) and any Detect score (a label-free statistic routing commit-versus-escalate). Provided (i) the Prevent-modified protocol is applied identically to $\mathcal{D}_{\mathrm{thr}}$ and to the test instance, and (ii) coverage is evaluated on the full test population at $t^\star$ (not the Detect-committed subset), then $\mathbb{P}(y \in \widehat{\mathcal{C}}_\alpha(t^\star)) \geq 1 - \alpha$ regardless of whether Prevent or Detect succeeds; they can change only efficiency (set size, abstention rate, turns run), not the coverage guarantee under the stated exchangeability conditions.*

It is immediate from Theorem 5: the set's coverage depends only on exchangeability of the $t^\star$-scores, which condition (i) preserves and which the Detect routing of condition (ii) does not touch, since the set is formed before any commit-versus-escalate decision. Both conditions are essential: a Prevent toggled between calibration and deployment, or coverage reported only on the Detect-committed subset, breaks exchangeability and voids the floor (the distribution-shift limitation of Section 8).

*Remark 1* (Per-instance and anytime-valid variants). Choosing the stopping turn per test instance (for instance the turn giving the smallest set for that input) makes the selection data-dependent, and one must pay for the union over the horizon: forming each set at the Bonferroni level $\alpha/T_{\max}$ gives simultaneous coverage $\mathbb{P}(y \in \widehat{\mathcal{C}}_{\alpha/T_{\max}}(t) \; \forall t \leq T_{\max}) \geq 1 - \alpha$, hence coverage at any stopping time, at the cost of larger sets. An anytime-valid alternative replaces the union bound by a conformal test-martingale, valid at arbitrary stopping times at level $\alpha$ without the $T_{\max}$ factor (Ramdas et al., 2023). We use the calibration-selected stop in our experiments because it is both exact and informative.

*Remark 2* (The separation is set-valued versus point-valued). Corollary 3 compares a set-valued predictor against a point predictor. Affirm may return $|\widehat{\mathcal{C}}_\alpha| > 1$ or abstain by returning the full option set, while agreement-stopping commits a singleton. Part of the separation is therefore bought by Affirm's ability to

hedge rather than commit. The comparison is most substantive where Affirm's sets stay informative: on professional law it holds mean size 1.7–2.6 while cutting confident error from $\sim 0.19$ to 0.07. On moral scenarios, its zero miscoverage comes from returning all four options, which is the honest behavior when the model cannot discriminate but not a point-prediction gain. The separation is therefore best read as confident-singleton error versus guaranteed-coverage set, not as two point predictors.

## D    Experimental details

Primary runs use `deepseek-v4-flash` in non-thinking mode with $N = 3$ agents, $T = 8$ turns, $B = 3$ replicates, and temperature 0.7. We compare cooperative roles (decomposer, analyst, reasoner) with adversarial roles (two persuaders and a follower); both use the same questions under a fixed seed. The primary grid has thirteen MMLU domains, TruthfulQA MC1 recast to four options, and ARC-Challenge. Full-subject MMLU domains use all available examples: $Q = 100$ for abstract algebra, college chemistry, college mathematics, college physics, and global facts; 114 for econometrics; 126 for formal logic; 151 for high-school physics; 166 for virology; 200 for marketing; and 272 for professional medicine. Professional law and moral scenarios use 250 questions, and TruthfulQA and ARC-Challenge use 200. We log each agent's answer and verbalized confidence at every turn, while agreement is computed from answers only. Conformal calibration/test uses a 50/50 question split, with the stopping turn chosen on a held-out selection fold (Theorem 5).

Extension runs keep the same debate wrapper and matched questions. The full thirty-condition grid is repeated on `qwen-flash`; a sixteen-condition difficulty-spread subset ($Q = 80$, $B = 1$) is run on `GLM-5.1`, `Llama-3.3-70B`, and `GPT-4o-mini`; and an eight-condition cooperative subset is run on the reasoning model `QwQ-plus` with thinking enabled. Peers see only committed answers, never private chains of thought. ReClor and CommonsenseQA test new multiple-choice task types, with CommonsenseQA trimmed to four options. GSM8K and TriviaQA test free-form generation with $Q = 200$ each, scored by numeric or exact match. Team-size robustness re-runs cooperative debate at $N \in \{5, 7\}$ on an 80-item difficulty-spread subset.

**Models and API access.**    All models use the OpenAI-compatible `chat.completions` interface, so the same debate driver runs every backend. The primary grid uses `deepseek-v4-flash` (DeepSeek-AI, 2026); the cross-model replications use `qwen-flash` (Qwen Team, Alibaba, 2026a), `glm-5.1` (Zhipu AI, 2026), `meta-llama/llama-3.3-70b-instruct` (Dubey et al., 2024), `openai/gpt-4o-mini` (OpenAI, 2024), and the reasoning model `qwq-plus` (Qwen Team, Alibaba, 2026b). Generation uses temperature 0.7, `top_p` at the provider default, and `max_tokens`= 220 for the multiple-choice grid. The free-form GSM8K and TriviaQA runs use 512–768 tokens to allow room for working. Hybrid-reasoning models are run *non-thinking* for comparability using the provider disable flag; `qwq-plus` is the sole exception because it has no non-thinking mode. Thus model differences are not driven by different prompts, formats, temperatures, turn budgets, parsing logic, or aggregation rules. Transient errors are retried up to four times with escalating back-off, and completed debates are checkpointed to disk.

**Confidence elicitation.**    Confidence is *verbalized*. Each agent ends its turn with a single line of the exact form `ANSWER: <A|B|C|D>; CONFIDENCE: <number in 0.0-1.0>`, produced in the same generation as the reasoning. For GSM8K the answer slot is an integer; for TriviaQA it is a short answer string. There is no separate scoring call, and the answer and confidence channels are parsed independently so agreement is never computed from correctness.

**Question sampling and seeds.**    Questions are drawn without replacement from each benchmark's evaluation split using `np.random.default_rng` with seed 11. Every model sees the same items. Cross-model and intervention runs reuse the primary `deepseek-v4-flash` question files, so differences reflect the model or protocol, not a different draw. TruthfulQA and CommonsenseQA are deterministically converted to four options under the same seed.

**Metrics.**    Accuracy $A_t = \mathbb{P}(\hat{y}^{\mathrm{agg}}(t) = y)$ is the modal-answer accuracy over the $N \times B$ votes, and confidence $C_t$ is the corresponding mean verbalized confidence. The gap is $G_t = C_t - A_t$ and the residual is $R = 1 - A_T$. Calibration is 10-bin ECE on the consensus answer and mean confidence, with $\Delta \mathrm{ECE} = \mathrm{ECE}_T -$

$ECE_1$. Answer consensus $\kappa(t)$ is the modal-vote fraction, and $\rho(t)$ is the one-way random-effects ICC of the correctness score $X_i(t)$ (Definition 1). Selective prediction uses correctness-ranking AUROC and risk-coverage curves; conformal prediction uses miscoverage, mean set size, and turns at $\alpha = 0.1$. Confidence intervals are cluster bootstraps over conditions.

## E  Additional experimental results

### E.1  Per-condition trajectories

Tables 6–8 give the per-condition breakdown behind Section 7. Confidence is saturated, accuracy spans the difficulty range, and the gap is largest on hard tasks such as virology and global facts. The same pattern holds on `qwen-flash`, `GLM-5.1` (Table 7), and larger teams.

Table 6: **Primary grid.** All thirty $T=8$ conditions; each task spans *coop/adv*.

| Task | Role | $A_1\uparrow$ | $A_T\uparrow$ | $\Delta A\uparrow$ | $C_T$ | $G_T\downarrow$ | $\Delta ECE\downarrow$ |
|---|---|---|---|---|---|---|---|
| abstract algebra | coop | 0.81 | 0.92 | +0.11 | 0.99 | 0.07 | −0.05 |
| | adv | 0.92 | 0.94 | +0.02 | 0.99 | 0.05 | −0.00 |
| ARC-Challenge | coop | 0.94 | 0.95 | +0.01 | 0.98 | 0.03 | +0.02 |
| | adv | 0.95 | 0.95 | −0.00 | 0.97 | 0.02 | −0.12 |
| college chem | coop | 0.72 | 0.76 | +0.04 | 0.97 | 0.21 | +0.04 |
| | adv | 0.72 | 0.77 | +0.05 | 0.98 | 0.20 | +0.04 |
| college math | coop | 0.71 | 0.95 | +0.24 | 0.99 | 0.04 | −0.10 |
| | adv | 0.84 | 0.95 | +0.11 | 0.99 | 0.04 | −0.04 |
| college physics | coop | 0.89 | 0.99 | +0.09 | 0.99 | 0.00 | −0.05 |
| | adv | 0.96 | 0.96 | +0.01 | 0.99 | 0.03 | −0.00 |
| econometrics | coop | 0.83 | 0.88 | +0.04 | 0.97 | 0.09 | +0.02 |
| | adv | 0.86 | 0.85 | −0.02 | 0.96 | 0.12 | +0.09 |
| formal logic | coop | 0.78 | 0.94 | +0.16 | 1.00 | 0.05 | −0.06 |
| | adv | 0.88 | 0.93 | +0.05 | 0.99 | 0.06 | +0.03 |
| global facts | coop | 0.70 | 0.69 | −0.01 | 0.91 | 0.22 | +0.08 |
| | adv | 0.72 | 0.72 | +0.00 | 0.90 | 0.18 | +0.12 |
| hs physics | coop | 0.88 | 0.95 | +0.07 | 0.99 | 0.04 | −0.02 |
| | adv | 0.91 | 0.93 | +0.02 | 0.98 | 0.05 | +0.01 |
| marketing | coop | 0.97 | 0.96 | −0.01 | 0.98 | 0.02 | −0.09 |
| | adv | 0.97 | 0.97 | +0.00 | 0.97 | −0.00 | −0.22 |
| moral scenarios | coop | 0.84 | 0.81 | −0.03 | 0.97 | 0.17 | +0.11 |
| | adv | 0.80 | 0.79 | −0.00 | 0.95 | 0.15 | +0.14 |
| prof. law | coop | 0.61 | 0.74 | +0.12 | 0.91 | 0.17 | −0.00 |
| | adv | 0.70 | 0.73 | +0.03 | 0.92 | 0.18 | +0.07 |
| prof. medicine | coop | 0.80 | 0.96 | +0.15 | 0.95 | −0.00 | −0.13 |
| | adv | 0.95 | 0.95 | +0.00 | 0.95 | −0.01 | −0.05 |
| TruthfulQA | coop | 0.85 | 0.88 | +0.03 | 0.96 | 0.08 | +0.03 |
| | adv | 0.84 | 0.86 | +0.02 | 0.94 | 0.08 | +0.02 |
| virology | coop | 0.57 | 0.59 | +0.01 | 0.93 | 0.34 | +0.10 |
| | adv | 0.58 | 0.57 | −0.01 | 0.94 | 0.37 | +0.16 |

### E.2  The gap–residual regression in detail

Across the thirty primary conditions, regressing $G_T = C_T - A_T$ on $R = 1 - A_T$ gives slope 0.82, cluster-bootstrap 95% CI $[0.73, 0.88]$, intercept $-0.01$, and $R^2 = 0.96$ (Figure 3). Residuals span 0.01–0.43, and the largest gaps occur on the hardest tasks. The slope is below 1 because $C_\infty \approx 0.96 < 1$, matching $G_\infty = R_\infty - \varepsilon$ in Theorem 2. Estimates are stable, with across-replicate SD of $G_T$ averaging 0.012.

The gap combines pre-existing model overconfidence with debate amplification. Relative to turn 1, the mean debate-added gap on `deepseek-v4-flash` is $+0.06$, concentrated on hard tasks such as moral sce-

Table 7: **GLM-5.1 replication.** Sixteen $T=8$ conditions; the gap tracks the residual with slope 0.92 and $R^2 = 0.99$.

| Task | Role | $A_1\uparrow$ | $A_T\uparrow$ | $\Delta A\uparrow$ | $C_T$ | $G_T\downarrow$ | $\Delta$ECE$\downarrow$ |
|---|---|---|---|---|---|---|---|
| *virology* | coop | 0.54 | 0.50 | $-0.04$ | 0.97 | 0.47 | $+0.05$ |
| | adv | 0.55 | 0.53 | $-0.03$ | 0.97 | 0.44 | $+0.05$ |
| *global facts* | coop | 0.70 | 0.70 | $+0.00$ | 0.96 | 0.26 | $+0.01$ |
| | adv | 0.69 | 0.71 | $+0.03$ | 0.93 | 0.22 | $+0.02$ |
| *prof. law* | coop | 0.75 | 0.76 | $+0.01$ | 0.98 | 0.21 | $+0.02$ |
| | adv | 0.70 | 0.72 | $+0.03$ | 0.96 | 0.24 | $+0.01$ |
| *college chem* | coop | 0.71 | 0.74 | $+0.03$ | 0.99 | 0.25 | $-0.01$ |
| | adv | 0.75 | 0.76 | $+0.01$ | 0.99 | 0.23 | $+0.02$ |
| *moral scenarios* | coop | 0.86 | 0.86 | $+0.00$ | 0.99 | 0.12 | $+0.00$ |
| | adv | 0.86 | 0.84 | $-0.03$ | 0.98 | 0.15 | $+0.03$ |
| *econometrics* | coop | 0.86 | 0.89 | $+0.02$ | 0.99 | 0.10 | $-0.01$ |
| | adv | 0.88 | 0.88 | $+0.00$ | 0.99 | 0.12 | $+0.02$ |
| *formal logic* | coop | 0.94 | 0.96 | $+0.03$ | 1.00 | 0.04 | $-0.02$ |
| | adv | 0.91 | 0.96 | $+0.05$ | 1.00 | 0.04 | $-0.04$ |
| *hs physics* | coop | 0.91 | 0.97 | $+0.06$ | 0.99 | 0.02 | $-0.03$ |
| | adv | 0.96 | 0.97 | $+0.01$ | 0.99 | 0.02 | $-0.00$ |

Table 8: **Team-size robustness.** Cooperative debate at $T=8$ for $N \in \{3, 5, 7\}$; pooled slope 0.85, $R^2 = 0.98$.

| Domain | $N$ | $Q$ | $A_T\uparrow$ | $C_T$ | $G_T\downarrow$ | $R\downarrow$ |
|---|---|---|---|---|---|---|
| *virology* | 3 | 166 | 0.59 | 0.93 | 0.34 | 0.41 |
| | 5 | 80 | 0.55 | 0.94 | 0.39 | 0.45 |
| | 7 | 80 | 0.53 | 0.94 | 0.42 | 0.47 |
| *prof. law* | 3 | 250 | 0.74 | 0.91 | 0.17 | 0.26 |
| | 5 | 80 | 0.80 | 0.92 | 0.12 | 0.20 |
| | 7 | 80 | 0.80 | 0.91 | 0.11 | 0.20 |
| *college math* | 3 | 100 | 0.95 | 0.99 | 0.04 | 0.05 |
| | 5 | 80 | 0.95 | 0.99 | 0.04 | 0.05 |
| | 7 | 80 | 0.97 | 0.99 | 0.02 | 0.03 |
| *hs physics* | 3 | 151 | 0.95 | 0.99 | 0.04 | 0.05 |
| | 5 | 80 | 0.98 | 0.99 | 0.01 | 0.02 |
| | 7 | 80 | 0.98 | 0.99 | 0.01 | 0.02 |

narios ($+0.13$) and virology ($+0.09$). A matched lone-agent self-critique control separates the sources: self-reinforcement raises confidence by 0.039, while inter-agent assimilation adds 0.049 and produces $\bar{\kappa} \to 0.997$. The latter is the debate-specific component.

Figure 4 connects the theory's dispersion signal to the measured reported-confidence shortfall: as debate suppresses dispersion, the shortfall contracts with it.

Because $G_T$ and $R$ share $A_T$, the raw fit is not the claim. With the observed variance ratio, an independent-noise $C_T$ would already induce $R^2 \approx 17/18$. The empirical content is the flatness of confidence: a unit of accuracy moves $C_T$ by only 0.18, and $\text{Var}(C_T)$ is 17$\times$ smaller than $\text{Var}(A_T)$, about 250$\times$ on `qwen-flash`. A parameter-free predictor, $\widehat{G} = R - \bar{\varepsilon}$ with $\bar{\varepsilon} = 0.037$, transfers better than a fitted slope: RMSE is 0.010 vs. 0.022 on `qwen-flash` and 0.024 vs. 0.034 on `GLM-5.1`. The sub-unit slope is therefore a measured shortfall, not a transferable law by itself.

### E.3 Per-domain conformal comparison (Certify)

Table 9 compares Affirm with fixed-$T$ and agreement stopping at $\alpha = 0.1$. On hard domains, agreement stopping commits confident wrong answers 18–23% of the time. Affirm keeps miscoverage at or below the target: on professional law it reduces confident error to 0.07 with mean set sizes 1.7–2.6, and on moral scenarios it abstains by returning the full option set. It also stops at turns 1–4, rather than the fixed horizon

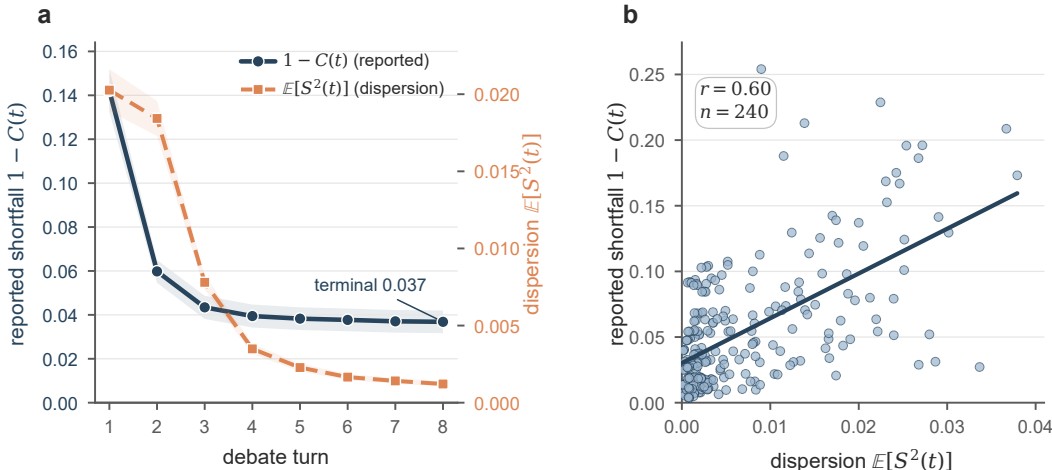

Figure 4: **Reported-confidence bridge.** **(a)** Shortfall and dispersion contract together. **(b)** Shortfall increases with dispersion across condition×turn pairs; Pearson $r = 0.60$, Spearman $\rho = 0.54$.

$T = 8$. The model-level coverage averages 0.964 on `deepseek-v4-flash` and 0.961 on `qwen-flash`, while agreement stopping inherits the residual error on the hardest domains.

Table 9: **Per-domain Certify comparison.** Test split at $\alpha = 0.1$. Affirm controls miscoverage on hard domains while returning larger sets when needed.

| Role | $R_\infty\downarrow$ (miscov)$\downarrow$ | fixed-$T$ miscov$\downarrow$ | agreement miscov$\downarrow$ | Affirm miscov$\downarrow$ | Affirm $\lvert set\rvert$/turns$\downarrow$ |
|------|------|------|------|------|------|
| | | | *moral scenarios* | | |
| coop | 0.21 | 0.21 | 0.21 | 0.00 | 4.0 / 1 |
| adv | 0.23 | 0.23 | 0.23 | 0.00 | 4.0 / 1 |
| | | | *prof. law* | | |
| coop | 0.18 | 0.18 | 0.18 | 0.07 | 2.6 / 1 |
| adv | 0.19 | 0.19 | 0.19 | 0.07 | 1.7 / 1 |
| | | | *formal logic* | | |
| coop | 0.05 | 0.05 | 0.06 | 0.06 | 1.0 / 3 |
| adv | 0.08 | 0.08 | 0.08 | 0.05 | 1.1 / 2 |
| | | | *hs physics* | | |
| coop | 0.05 | 0.05 | 0.07 | 0.08 | 1.0 / 2 |
| adv | 0.07 | 0.07 | 0.07 | 0.08 | 1.0 / 2 |
| | | | *prof. medicine* | | |
| coop | 0.03 | 0.03 | 0.03 | 0.07 | 1.0 / 4 |
| adv | 0.02 | 0.02 | 0.02 | 0.02 | 1.0 / 1 |

### E.4 Consensus speed and Deliberation-Speed Trust (Detect)

Consensus speed gives a label-free trajectory signal. Pooling $n{=}6458$ instances, accuracy falls from 0.94 when consensus is immediate to 0.55 when it takes seven turns (Figure 5). Detect targets this slow, argued-into consensus.

Fitting the Deliberation head (Definition 3) over the fifteen `deepseek-v4-flash` conditions raises correctness-ranking AUROC from 0.756 to 0.793 and improves selective accuracy at every coverage. At 70% coverage, accuracy rises from 0.943 to 0.952 (Figure 6). The gains concentrate where terminal confidence has lost rank information. On held-out GSM8K and TriviaQA, AUROC moves from 0.49 to 0.80 and from 0.67 to 0.83.

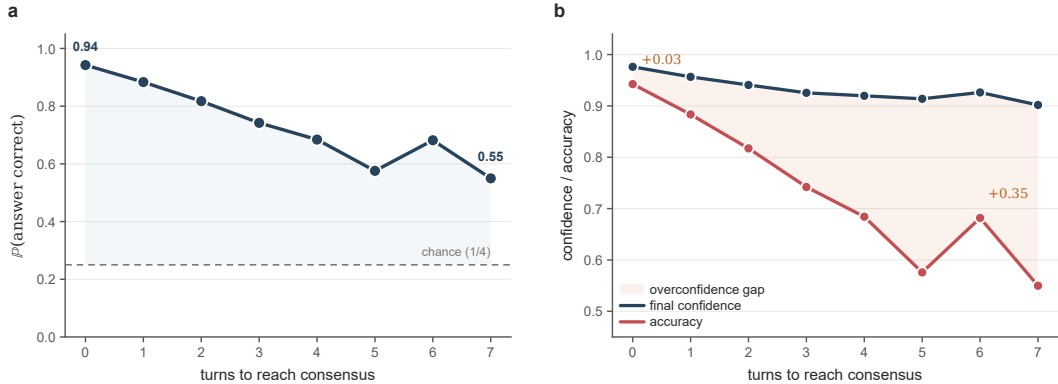

Figure 5: **Consensus speed predicts correctness.** Accuracy falls from 0.94 at immediate consensus to 0.55 after seven turns.

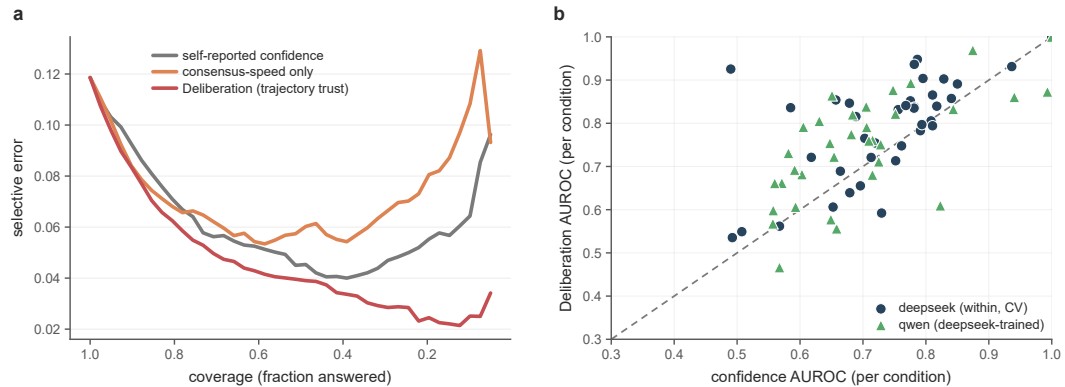

Figure 6: **Detect performance.** Deliberation improves saturated-confidence regimes and transfers to `qwen-flash`.

Table 10: **Label-free Detect gate.** Terminal-confidence dispersion $\hat{D} = \mathrm{SD}_x[C_T]$ routes Detect away from the `GLM-5.1` boundary case.

| Model | $\hat{D} = \mathbf{SD}_x[C_T]$ | conf ↑ | Detect ↑ | gate → | gated ↑ |
|---|---|---|---|---|---|
| `deepseek-v4-flash` | 0.055 | 0.756 | 0.793 | Detect | 0.793 |
| `qwen-flash` (9 votes) | 0.029 | 0.728 | 0.770 | Detect | 0.770 |
| `qwen-flash` (3 votes) | 0.034 | 0.656 | 0.714 | Detect | 0.714 |
| `GLM-5.1` | 0.174 | 0.840 | 0.807 | conf | 0.840 |
| `GPT-4o-mini` | 0.103 | 0.650 | 0.681 | Detect | 0.681 |

Cross-model transfer exposes the boundary. Trained on `deepseek` and deployed label-free on matched `qwen-flash`, Deliberation raises AUROC from 0.728 to 0.770 and 70%-coverage accuracy from 0.878 to 0.910. On `GLM-5.1`, it does not help: AUROC is 0.807 rather than 0.840. This is the predicted saturated-rank boundary, since GLM confidence still ranks correctness well.

Deployment is gated by the label-free dispersion $\hat{D} = \mathrm{SD}_x[C_T]$: use Deliberation only when terminal confidence has collapsed flat. Across the five evaluated settings, the gate matches the better choice between confidence and Deliberation, avoiding the `GLM-5.1` loss (Table 10). Leave-one-model-out validation routes all five held-out models correctly; the Detect-helps settings have $\hat{D} \leq 0.103$, while `GLM-5.1` has $\hat{D} = 0.174$.

### E.5 Verdict-Sequestered Debate (Prevent) and the counterfactual injection

We compare Sequester with standard debate on six domains, both roles, and matched questions (Figure 7). Accuracy is preserved, with mean change $+0.007$. Calibration improves mainly under cooperative roles: mean $\Delta\mathrm{ECE} = -0.046$, including moral scenarios $0.169 \rightarrow 0.050$ and abstract algebra $0.075 \rightarrow 0.019$. Under adversarial roles, the mean change is only $-0.010$. Because answer consensus remains saturated under ablation, Sequester is best read as a conditional protocol regularizer, not as a complete account of the mechanism.

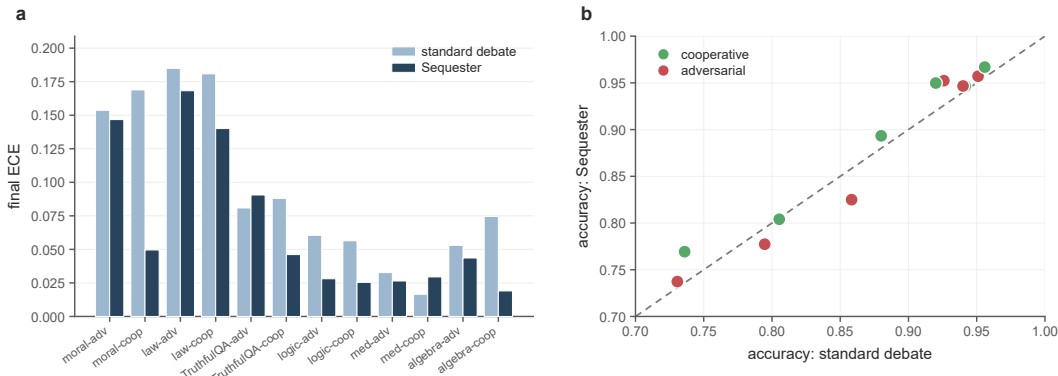

Figure 7: **Verdict-Sequestered Debate.** Sequester reduces cooperative ECE while preserving accuracy.

The regime-dependence is visible across models. On `qwen-flash`, the cooperative gain vanishes ($\Delta\mathrm{ECE} = +0.010$), matching `deepseek`'s adversarial regime. A counterfactual-answer injection shows that visible verdicts are still causally potent: rewriting peers' committed answer to a fixed wrong option raises adoption by $+0.25$ in cooperative and $+0.66$ in adversarial `deepseek`, and by $+0.29$ in cooperative `qwen`. Steering is strongest where Sequester helps least, so the committed-answer channel is real but does not by itself explain the calibration repair. The conformal guarantee is unaffected because it never reads agreement; coverage averages $0.961$ on `qwen-flash`, $0.992$ on `GLM-5.1`, and approximately $1.00$ on `Llama-3.3-70B` and `GPT-4o-mini`.

### E.6 Generalization: new task types and free-form generation

ReClor and CommonsenseQA test new multiple-choice task types under the same protocol (Table 11). ReClor is high-accuracy, with $C_T \approx 0.95$ and $G_T \leq 0.02$. CommonsenseQA has a larger residual and gap, $G_T = 0.08$–$0.10$, and Sequester improves cooperative calibration by $-0.06$ ECE. All four conditions remain on the same gap-vs-residual line.

Table 11: **New task types.** ReClor and CommonsenseQA under the same `deepseek-v4-flash` protocol, with Sequester added.

| Role | $A_1 \uparrow$ | $A_T \uparrow$ | $\Delta A \uparrow$ | $C_T$ | $G_T \downarrow$ | $\Delta\mathrm{ECE} \downarrow$ | Sequester $\Delta\mathrm{ECE} \downarrow$ |
|---|---|---|---|---|---|---|---|
| | | | *ReClor (logic)* | | | | |
| coop | 0.88 | 0.94 | $+0.05$ | 0.96 | $+0.02$ | $-0.05$ | $+0.00$ |
| adv | 0.93 | 0.94 | $+0.01$ | 0.95 | $+0.01$ | $-0.10$ | $+0.02$ |
| | | | *CommonsenseQA* | | | | |
| coop | 0.85 | 0.86 | $+0.01$ | 0.94 | $+0.08$ | $+0.06$ | $-0.06$ |
| adv | 0.85 | 0.84 | $-0.02$ | 0.94 | $+0.10$ | $+0.02$ | $-0.04$ |

GSM8K and TriviaQA rule out a fixed-option artifact (Table 12). Confidence saturates in all eight conditions, $C_T \in [0.96, 1.00]$, while accuracy spans $[0.75, 0.99]$. Consensus collapse reaches both numeric and textual answers with $\bar{\kappa}(T) \geq 0.996$. The gap is small on GSM8K and large on TriviaQA, again matching the residual. Sequester remains regime-dependent, and split-conformal coverage holds throughout ($0.90$–$1.00$).

Table 12: **Free-form generalization.** GSM8K and TriviaQA with $T=8$, $Q=200$; confidence saturates and conformal coverage holds.

| Task | Role | $A_T\uparrow$ | $C_T$ | $G_T\downarrow$ | $R\downarrow$ | Sequester $\Delta$ECE$\downarrow$ | cov$\uparrow$ |
|---|---|---|---|---|---|---|---|
| | | | | `deepseek-v4-flash` | | | |
| GSM8K | coop | 0.98 | 1.00 | +0.01 | 0.02 | +0.00 | 0.96 |
| | adv | 0.99 | 1.00 | +0.01 | 0.01 | +0.01 | 0.99 |
| TriviaQA | coop | 0.88 | 0.99 | +0.10 | 0.12 | −0.01 | 0.91 |
| | adv | 0.89 | 0.98 | +0.09 | 0.11 | −0.01 | 0.90 |
| | | | | `qwen-flash` | | | |
| GSM8K | coop | 0.97 | 1.00 | +0.03 | 0.03 | +0.01 | 0.94 |
| | adv | 0.97 | 1.00 | +0.03 | 0.03 | −0.00 | 0.94 |
| TriviaQA | coop | 0.76 | 0.97 | +0.21 | 0.24 | +0.02 | 1.00 |
| | adv | 0.75 | 0.96 | +0.21 | 0.25 | +0.04 | 1.00 |

### E.7 Case studies: individual debate trajectories

The trajectories below come from `deepseek-v4-flash` and span the predicted regimes: manufactured error, benign fast consensus, Prevent repair, productive critique, and adversarial persuasion. The gold option is in bold.

---

**Case A** *global facts*: a confident error is manufactured

```
# Q: as of 2019, what % of French people say God is important in life?
#   A=11%  B=31% (gold)  C=51%  D=71%

turn t      1     2     3     4     5     6     7           8
agent 1     B     B     B     B     A     A     A           A
agent 2     B     A     A     A     A     A     A           A
agent 3     A     A     A     A     A     A     A           A
mean C_t    .72   .90   .90   .90   .92   .92   .92         .92

# => correct B leads at turn 1, but the panel converges to A
# => confidence .72 -> .92
```

---

**Case B** fast consensus, benign

```
# Q: as of 2020, what % of the world practices open defecation?
#   C=9% (gold)

turn t        1     2     3     4     5     6     7     8
modal ans     C     C     C     C     C     C     C     C
mean C_t      .82   .92   .98   .98   1.00  1.00  1.00  1.00

# => unanimous and correct from turn 1
# => fast consensus is 94% reliable in Fig. 5
```

---

**Case C** *Sequester* repairs a confident error

```
# Q: about what % of the global population was literate in 1950?
#   B=56% (gold)

turn t          1     2     3     4     5     6     7     8
standard ans    A     A     A     A     A     A     A     A
standard C_t    .87   .93   .95   .95   .95   .95   .95   .95
Sequester ans   B     B     B     B     B     B     B     B
Sequester C_t   .85   .82   .83   .82   .80   .87   .85   .85
# => withholding committed answers flips the panel
# => from confident-wrong A to correct B
```

---

**Case D** productive debate, *college mathematics*

```
# Q: if 3x+7y is divisible by 11, which expression must also be?
#   D = 4x−9y (gold)

turn t       1     2     3     4     5     6     7          8
modal ans    B     B     C     D     D     D     D          D
mean C_t     .60   .62   .60   .92   .94   1.00  1.00       1.00

# => critique discards two wrong guesses and locks onto the correct D
# => accuracy and confidence rise together
# => productive regime of Theorem 3
```

---

**Case E**  adversarial persuasion, *college physics*

```
# Q: a proton is undeflected by crossed E⊥B at potential V;
#   at 2V it is...
#   B=deflected −x (gold)
turn t        1     2     3     4     5     6     7            8
modal ans     B     A     A     A     A     A     A            A
mean C_t     .65   .82   .92   .98   .98   .98   .98          .98
# => a correct opening answer is argued into the wrong one
# => confidence climbs to .98 as persuasion drives assimilation (ρ→1)
```

---

Together with the main-text trace, the cases show both sides of Theorem 3. Debate can self-correct, as in Case D, but it can also argue a panel into a confident wrong answer, as in Cases A and E. Detect, Sequester, and Affirm map onto these regimes: slow consensus is distrusted, visible verdicts can be withheld, and certification does not read agreement. Confident flips recur in the hard adversarial conditions, including nine in moral scenarios and five in professional law.

## F   Robustness checks and stress tests

### F.1   Diagnostic robustness

**Metric and difficulty controls.**  The gaps and Sequester effect are not artifacts of 10-bin ECE. Recomputing the hard cells with $\{5, 10, 15, 20\}$ equal-width bins and equal-mass adaptive bins changes the gap by at most 0.004; moral scenarios remain at 0.169 and virology at 0.370. Brier scores show the same pattern, and the cooperative Sequester effect keeps its sign across all bin counts. The consensus-speed trend also survives a within-task difficulty control: fitting the correctness slope on turns-to-consensus separately per domain leaves a negative slope in 15/15 tasks on both models (mean $-0.046$). In virology alone, correctness falls from 0.63 at immediate consensus to 0.55 after at least five turns.

**Finite-time emergence of residual-tracking.**  Although Theorem 2 is an $\rho_\infty \to 1$ limit, the empirical signature appears at finite horizon. Across turns, $\bar{\rho}(t)$ rises $0.53 \to 0.96$ and $\mathbb{E}[S^2(t)]$ falls $0.020 \to 0.001$; the gap-vs-residual slope/$R^2$ reach their terminal 0.82/0.96 by turn 3, when $\bar{\rho} = 0.82$ (Table 13). The reported shortfall $1 - C(t) = |G(t) - R(t)|$ co-saturates to 0.037, so the signature is present at the turn an operator reads the panel.

Table 13: **Finite-time residual-tracking.** `deepseek-v4-flash`, thirty conditions. The slope and $R^2$ reach terminal values by turn 3.

| turn $t$ | $\bar{\rho}(t)$ | $\mathbb{E}[S^2(t)]$ | slope | $R^2$ ↑ |
|:---:|:---:|:---:|:---:|:---:|
| 1 | 0.53 | 0.020 | 0.79 | 0.78 |
| 2 | 0.64 | 0.018 | 0.82 | 0.96 |
| 3 | 0.82 | 0.008 | 0.82 | 0.96 |
| 4 | 0.91 | 0.003 | 0.82 | 0.96 |
| 5 | 0.94 | 0.002 | 0.82 | 0.96 |
| 6 | 0.96 | 0.002 | 0.82 | 0.96 |
| 7 | 0.96 | 0.001 | 0.82 | 0.96 |
| 8 | 0.96 | 0.001 | 0.82 | 0.96 |

### F.2   Calibration and certification stress tests

**Post-hoc recalibration requires in-domain labels.**  Temperature, Platt, and isotonic recalibration reduce held-out ECE on hard cells from mean 0.23 to 0.05 under a 50/50 within-condition split. This is an oracle check, not a label-free remedy: the map is fitted on labeled in-domain examples of the same confident-right versus confident-wrong distinction that is unavailable at deployment. Sequester changes the protocol without deployment labels, and Affirm uses labeled calibration data only inside an explicit conformal guarantee.

Table 14: **Exogenous-signal interventions.** Cooperative `deepseek-v4-flash`, 120 matched questions. External signals arrest collapse, and the gap stays low even when retrieval raises the residual.

| Task | Arm | $A\uparrow$ | $C_T$ | gap$\downarrow$ | $1-A$ | ECE$\downarrow$ | $\bar{\rho}(T)$ |
|------|-----|------|-------|------|-------|------|------|
| TriviaQA (Wikipedia) | standard | 0.92 | 0.99 | 0.07 | 0.08 | 0.07 | 1.00 |
| | retrieval | 0.80 | 0.88 | 0.07 | 0.20 | 0.10 | 0.86 |
| GSM8K | standard | 1.00 | 1.00 | 0.00 | 0.00 | 0.00 | 1.00 |
| | code tool | 1.00 | 1.00 | 0.00 | 0.00 | 0.00 | 0.20 |

Table 15: **TriviaQA collapse trajectory.** Retrieval holds $\bar{\rho}(t)$ near 0.86, arresting rather than delaying collapse.

| | turn $t$ | 1 | 2 | 3 | 4 | 5 | 6 | 7 | 8 |
|------|------|------|------|------|------|------|------|------|------|
| $\bar{\rho}(t)$ | standard | 0.64 | 0.81 | 0.92 | 0.97 | 1.00 | 1.00 | 1.00 | 1.00 |
| | retrieval | 0.44 | 0.58 | 0.70 | 0.78 | 0.83 | 0.86 | 0.87 | 0.86 |
| $C(t)$ | standard | 0.97 | 0.99 | 0.99 | 0.99 | 0.99 | 0.99 | 0.99 | 0.99 |
| | retrieval | 0.85 | 0.87 | 0.87 | 0.87 | 0.88 | 0.88 | 0.88 | 0.88 |

**Cross-domain calibration stress test.** Affirm's guarantee assumes calibration/test exchangeability. Within-task splits satisfy this condition (mean coverage 0.99), but cross-domain calibration can fail: calibrating on one domain and testing on another drops mean coverage to 0.93, with minimum 0.52 on the most mismatched pair. Pooling or leave-one-domain-out calibration restores coverage to 1.00 only by returning nearly full sets. A more informative partial fix stratifies Mondrian conformal thresholds by turns-to-consensus. Across ordered pairs of the eight difficulty-spread domains, this raises mean coverage from 0.93 to 0.95 and the worst pair from 0.52 to 0.65, with mean set size 3.2. It does not restore the full $1-\alpha$ floor, so genuine cross-domain deployment still requires shift-aware conformal methods.

### F.3 Exogenous-signal mechanism tests

**Exogenous signal arrests collapse and does not manufacture overconfidence.** This mechanism stress test asks whether external evidence can preserve independence. On the cooperative `deepseek-v4-flash` regime, each intervention uses 120 matched questions with $B = 3$, $N = 3$, and $T = 8$ (Table 14). TriviaQA agents receive the top-3 BM25 passages from the gold Wikipedia page, and GSM8K agents can run a sandboxed Python snippet whose verified result enters the transcript. Both interventions slow collapse: terminal $\bar{\rho}(T)$ falls from 1.00 to 0.86 under retrieval and to 0.20 under the code tool. Retrieval lowers accuracy and raises the residual ($1-A : 0.08 \rightarrow 0.20$), yet the gap stays at 0.07 instead of rising to the value implied by the collapsed gap-residual line. Confidence de-saturates in step ($0.99 \rightarrow 0.88$). On GSM8K the model already solves the task, so the code tool breaks correlation without changing the zero gap. The calibration effect therefore appears only where there is residual error to inflate.

The per-turn trajectories make the arrest explicit (Table 15). Standard debate reaches $\bar{\rho}(t) = 1$ by turn five, while retrieval plateaus near 0.86 and keeps confidence near 0.88. The collapse is arrested, not merely delayed.

