# OpenReview forum: "When Consensus Is Not Correctness: Diversity Collapse and Manufactured Overconfidence in Multi-Agent LLM Debate"
_TMLR — Under review for TMLR_

### Review · Reviewer_XJ2S · 2026-06-20

**Summary Of Contributions:**

This paper studies whether consensus in multi-agent LLM debate can be trusted as a signal of correctness. The main finding is that debate may create manufactured overconfidence: agents become increasingly aligned through interaction, so final agreement and reported confidence can be high even when the answer is wrong. The paper formalizes this failure mode through a theoretical framing based on inter-agent correlation, reduced diversity, and calibration gap, and provides empirical evidence that consensus and confidence can saturate while accuracy still varies substantially across tasks. To address this issue, the paper proposes CMAD, a framework with Prevent, Detect, and Certify components. In particular, the Certify/Affirm component uses conformal prediction to return a calibrated answer set rather than always committing to the majority answer. Experiments show that this approach achieves lower miscoverage than the tested majority-voting, agreement-based, and confidence-based stopping baselines, especially on hard conditions. Overall, the paper is valuable because it shifts the focus from simply improving final-answer accuracy to understanding when multi-agent debate should or should not be trusted.

**Audience:**

Yes

**Audience Explanation:**

This work is very relevant to multi-agent systems, and useful for general ensemble learning.

**Broader Impact Concerns:**

There is no ethical concerns of this paper.

**Claims And Evidence:**

No

**Claims Explanation:**

## Strength

**S1. The paper studies an important problem beyond benchmark accuracy.** The paper focuses on whether multi-agent debate can correctly know when it does not know, instead of only improving final-answer accuracy. This is an important and timely problem, because many agent systems use agreement, confidence, or stability as stopping signals. The paper shows that these signals can be misleading when consensus is produced by the debate process itself. This makes the problem setting practically relevant and different from papers that only aim to improve benchmark scores.

**S2. The paper provides a useful theoretical framing for debate-induced overconfidence.** The paper gives a set of theoretical arguments to describe why multi-agent debate can lose diversity and produce misleading agreement. In particular, it connects inter-agent correlation, reduced disagreement, apparent confidence, and overconfidence gap. Although some results rely on assumptions, the analysis is still useful because it gives a clear language to describe the failure mode. It helps explain why consensus should not be directly interpreted as correctness.

**S3. The proposed CMAD framework is practically useful, especially the Certify component.** The paper proposes a clear framework with Prevent, Detect, and Certify components. Among them, Certify is the most convincing part, because it uses conformal prediction to replace agreement-based stopping with a coverage-oriented set prediction. This is a useful safety mechanism: instead of always committing to the majority answer, the method can return a larger answer set when the debate signal is not reliable. The experiments show that this component achieves lower miscoverage than the tested majority-voting, agreement-based, and confidence-based baselines, which supports its practical value.

## Weakness

**W1. The link from $S^2(t)$ to observable agreement is not clearly justified.** Theorem 1 is mathematically valid under its assumptions, but the proof is based on the latent quantity $S^2(t)$, which is defined using a label-dependent correctness score. This quantity is not directly observable during deployment. In contrast, the paper’s practical claim is about answer agreement or apparent confidence. Therefore, the theorem does not directly prove that debate will produce observable consensus; it assumes that the decrease of latent dispersion will also appear as reduced answer disagreement. This connection should be stated more clearly and supported empirically.

**W2. The existence of $C_\infty$, $A_\infty$, and $G_\infty$ needs stronger experimental evidence.** Theorem 2 relies on the idea that debate leads to stable terminal values of $C$, $A$, and $G$, and that the confidence shortfall $1-C$ is small and nearly constant. However, the paper does not clearly show that these values remain stable across different models, temperatures, or top-p settings. It is also unclear whether all debate trajectories reliably converge, or whether the final values depend strongly on the model and decoding configuration. The experiments show terminal behavior under the tested settings, but the paper seems to treat this observed behavior as a general convergence property. The authors should either provide stronger stability evidence across models and decoding settings, or state the claim more carefully as a finite-setting empirical observation.

**W3. Theorem 3 directly compares two quantities that may not be on the same scale.** Theorem 3 compares the increase in reported confidence, $\Delta C$, with the increase in accuracy, $\Delta A$, and concludes that the gap decreases when $\Delta A>\Delta C$. While this is algebraically correct from $G(t)=C(t)-A(t)$, the comparison assumes that changes in confidence and changes in accuracy are directly comparable. This assumption is not fully justified. Accuracy is an empirical correctness rate, while confidence is a raw verbalized score produced by the agents. Without calibration, a change of $0.1$ in confidence does not necessarily have the same meaning as a change of $0.1$ in accuracy. Therefore, Theorem 3 should not be presented as a strong theoretical result unless the confidence scale is calibrated or the claim is shown to be robust under reasonable transformations of confidence.

**W4. The comparison with agreement- and confidence-based baselines is not exhaustive.** The experiments show that Affirm achieves lower miscoverage than the tested stopping rules, such as `agreement@0.9` and `confidence@0.9`, on hard conditions. However, the paper does not report a systematic threshold sweep for agreement or confidence. Therefore, the results only support the claim that Affirm outperforms the selected representative baselines, not that it is better than every possible threshold choice. The stronger statement that no fixed agreement threshold is uniformly safe is theoretical and depends on the collapse or stabilized-answer setting. The paper should more clearly separate this conditional theoretical claim from the finite-threshold experimental comparison.

**Requested Changes:**

1. **Clarify the link between latent dispersion and observable agreement (W1).**

    The authors should explain more clearly why a decrease in $S^2(t)$ should imply reduced answer disagreement or higher apparent confidence. Since $S^2(t)$ is based on a label-dependent latent score, the paper should provide empirical evidence that this latent dispersion tracks observable agreement signals during debate.

2. **Provide stronger stability evidence or weaken the convergence-style claims (W2).**

    The authors should show whether the terminal values of $C$, $A$, and $G$ remain stable across different models, temperatures, and top-p settings, and whether debate trajectories reliably converge under longer horizons. If such stability is not established, the claims should be rewritten as finite-setting empirical observations rather than general convergence behavior.

3. **Justify the direct comparison between confidence and accuracy changes (W3).**

    The authors should either calibrate the reported confidence scale or show that the conclusions are robust under reasonable transformations of confidence. Without such evidence, the comparison between $\Delta C$ and $\Delta A$ should be presented more cautiously, since raw verbalized confidence and empirical accuracy may not be directly comparable.

4. **Strengthen the comparison with agreement- and confidence-based baselines (W4).**

    The authors should report systematic threshold sweeps or risk-coverage curves for agreement-based and confidence-based stopping rules. Otherwise, the empirical claim should be limited to the tested baselines, such as `agreement@0.9` and `confidence@0.9`, rather than implying superiority over all possible threshold choices.

---

### Review · Reviewer_jXpv · 2026-07-06

**Summary Of Contributions:**

The paper argues that agreement produced by multi-agent LLM debate is endogenous to the debate protocol and therefore cannot be used as a correctness or confidence signal. The contributions are threefold:

1. A "variance account" of debate built on the classical equicorrelation identity: as agents read each other, the inter-agent correlation rho(t) rises toward 1, which simultaneously destroys the averaging benefit of the ensemble (aggregate variance rises to the single-agent level) and drives observed disagreement to zero (agents look unanimous). Apparent confidence thus saturates independently of accuracy, and the terminal calibration gap equals the residual error minus a confidence shortfall ("manufactured overconfidence", Theorems 1-3).

2. An empirical validation of this signature: across thirty conditions (15 tasks x cooperative/adversarial roles) on deepseek-v4-flash, plus replications on five further model families, confidence saturates into [0.90, 1.0] while accuracy spans 0.57-0.99, terminal confidence has ~17x smaller variance than accuracy, and the gap-vs-residual regression is affine with slope 0.82 and R^2 = 0.96. Supporting controls include a matched lone-agent self-critique, a counterfactual verdict-injection probe, heterogeneous panels, team-size variation, and exogenous-signal tests (retrieval, code tool) that arrest the collapse.

3. Calibrated Multi-Agent Debate (CMAD), a framework with three levers: Prevent (withholding peers' committed verdicts), Detect (a trust score built from label-free trajectory features such as consensus speed), and Certify (a split-conformal set with marginal coverage guarantees that hold even under full diversity collapse), together with a stopping rule (Affirm) that provably keeps coverage while agreement-based stopping commits confident errors at 18-47% miscoverage.

Strengths:
- The mechanism is clearly articulated and tied to classical ensemble theory; the paper is honest about which parts are algebraic identities and which parts are empirical content (e.g., the co-saturation of reported confidence, the mechanical part of the R^2).
- The empirical work is thorough for the scope: many conditions, several model families, matched controls, and robustness/stress tests (recalibration oracle, cross-domain conformal failure, exogenous-signal tests).
- The conformal certification component is principled, and the safety separation from agreement-based stopping (Corollary 3) is a useful, quantified statement.

Weaknesses:
- The theoretical results are mostly elementary algebra or definitional identities (e.g., G = C - A rearrangements); the load-bearing empirical fact — that verbalized confidence co-saturates to a near-constant shortfall — is assumed/measured rather than derived.
- The qualitative conclusions (debate converges without necessarily being correct, LLMs are overconfident, agreement is not correctness) are largely in line with what prior work and practitioner experience already suggest; the formalization and the certification framing are the main new content.
- The benchmarks are older multiple-choice datasets (MMLU, ARC, TruthfulQA, GSM8K, TriviaQA), likely partly saturated or contaminated for current models. Moreover, these are single-shot QA tasks where nothing about the problem is inherently multi-turn or multi-agent; as evaluated, the setup is closer to an LLM ensemble with message passing than to a genuinely agentic setting, which weakens the "multi-agent" framing of the claims.
- Presentation issues: strong assumptions stated somewhat informally, forward references from Section 4 into Section 7 results, and a dense Figure 1.

**Additional Comments:**

In Table 1, all agreement/stability/budget stopping rules report identical error columns (miscoverage 0.22, etc.). I assume this is because consensus forms so early that every rule commits the same answer, but the caption should say so explicitly, as it otherwise looks like a copy-paste error.

**Audience:**

Yes

**Audience Explanation:**

Yes, but I would temper the level of interest. The central findings — that debate drives agents toward agreement regardless of correctness, that LLMs are overconfident, and that consensus is therefore a poor trust signal — are things much of the community already believed and that prior work has documented empirically (the paper itself cites Smit et al., Wynn et al., Kong et al., Liang et al., Huang et al.). What is new here is the formalization of the mechanism, the residual-tracking signature, and the certification-first framework with a conformal guarantee that survives collapse. That has some value for people building or gating multi-agent pipelines, so at least some of TMLR's audience would be interested, but the surprise value of the findings is limited.

**Broader Impact Concerns:**

I have no broader impact concerns. The work is aimed at making multi-agent LLM systems safer to deploy by flagging an unreliable trust signal and replacing it with a certified alternative; I do not see a need for a Broader Impact Statement.

**Claims And Evidence:**

Yes

**Claims Explanation:**

Mostly yes, within the scope the paper sets for itself. The theory is correct, though it is largely elementary: the equicorrelation identity is classical, and Theorems 2-3 are essentially rearrangements of the definition of the calibration gap. The empirical content is where the claims live, and there the evidence is reasonably convincing: the diversity collapse is measured directly (rho rising 0.53 to 0.96, dispersion vanishing), confidence saturation and the gap-vs-residual relation replicate across five model families with sub-unit slopes, and the paper includes the right controls (the lone-agent self-critique to separate baseline overconfidence from interaction-driven amplification, the counterfactual verdict injection, the heterogeneous panel, and the exogenous-signal tests showing that retrieval/tools arrest the collapse). The paper is also candid that the high R^2 is partly mechanical given the variance ratio, and offers the constant-shortfall out-of-sample prediction as the non-trivial content.

Caveats that keep this from being a strong yes: (i) the co-saturation of verbalized confidence — the bridge from the proven apparent-confidence result to the measured reported-confidence result — is an empirical regularity, not something the theory delivers, and the correlation supporting it is moderate (r = 0.60); (ii) the assumptions (e.g., bounded variance across all agents and turns, assimilative dynamics) are strong and stated somewhat informally; (iii) the evaluation uses older, partly saturated benchmarks, and these are single-shot QA tasks with no inherently multi-turn or agentic structure — so it is unclear how much headroom the "hard task" regime really probes for current models, and the evidence speaks more to LLM ensembles exchanging messages than to multi-agent systems in settings where agents are naturally deployed. None of these overturn the claims, but they should be addressed (see Requested Changes).

**Requested Changes:**

Experimental setup:

The datasets used for the experiments are rather old and should be updated to newer and harder versions.

Relatedly, the benchmarks are all single-shot QA tasks; none of them are settings that are natural for multi-turn or multi-agent operation (e.g., agentic tasks requiring tool use, planning, or interaction over multiple steps). As evaluated, the setup reads more like an LLM ensemble with message passing than a multi-agent system. Adding at least one benchmark where the multi-agent/multi-turn structure is intrinsic to the task would substantially strengthen the "agent" part of the claims. This, together with the dataset update above, is the change I consider critical; the remaining points below would strengthen the work.

Presentation:

Open with the “wisdom of the crowd” observation from populations of humans. This would help motivate the paper.

It would strengthen the paper if there were, early on (in Section 1 or 3), an example showcasing the problem with three problems: an easy, a medium, and a hard problem.

In Figure 1, if all four are below the true value in step 1, why would they safely land on the true value in step 3?

Make it very clear what the assumptions are. In Section 3, there should be a list of short and clear assumptions, and theorems/lemmas can then say “We assume 1, 2, 5, and 6” before stating the theorem/lemma.

The assumptions seem quite strong (e.g., bounded variance across all agents and time steps).

It is very weird that section 4 includes numbers from section 7. This does not make it very readable. As things are now, I would reorder the sections to have the results first.

Questions/things that are not clear:

What is the motivation behind the calibration gap G(t)? Firstly, why should the models be able to estimate their confidence, and secondly, why should this track the accuracy in a 1-to-1 ratio?

Minor writing issues:

The start of “Stopping rules and conformal certification” in the related work should be rewritten, as the flow is not good. Same for the first two paragraphs of Section 3.

---

### Review · Reviewer_ukHC · 2026-07-09

**Summary Of Contributions:**

I'm not familar with this area.  And to be honest, I can't understand the main contributions of this paper.  The paper proposes a theoretical model for multi-LLM debate.  Each LLM in each turn outputs an answer $\hat y_i(t)$ and a confidence $c_i(t)$.  Any two LLMs' answers are correlated according to a covariance $\rho(t)$.  Instead of the average reported confidence $C(t) = 1/N \sum_i c_i(t)$, an operator observes an "apparent confidence" $\tilde C(t)$, such as the fraction of votes on the modal answer, which equals 1 at unanimity.

* The first contribution of the paper seems to be showing the following theoretical result: in multi-LLM debate, when concensus is reached (models output the same answer), the LLM's confidence increases, but the accuracy of the common answer can still remain imperfect.
* Then, the paper introduces a framework called "Calibrated Multi-Agent Debate".  However, I can't understand the structure or the purpose of this framework.



**Strengths**: The topic is timely.  The motivation is clear.  It gives a good warning that multi-agent debate does not always work because the consensus reached from debate may result in overconfidence, not necessarily improving ansewr accuracy.



**Weaknessess**:

* The theoretical result is not a real contribution.  It is just a restatement of the assumption.  It is a standard fact that the variance of the aggregate answer among $N$ agents is $\frac{\sigma^2(t)}{N}[ 1 + (N-1) \rho(t)]$, where $\sigma^2(t)$ is the single agent variance and $\rho(t)$ is the two-agent covariance (Lemma 1).   If $\rho(t) \to 1$, then the aggregate variance converges to the single agent variance -- this is what the authors mean by "the advantage of debate vanishes".  However, instead of proving $\rho(t) \to 1$, the authors just assume in Assumption 1 that $\rho(t) \to 1$ as $t \to \infty$ as LLM agents debate more.  Then, of course the advantage of debate vanishes -- Theorem 1 is just a restatement of Assumption 1.  The key theoretical result to show here, in my opinion, is that $\rho(t)$ indeed converges $1$.

* The paper is poorly written.  Too many concepts are introduced without context.  Terms are used before being defined.  Some phrases are too vague.  The main contribution is not clearly presented.  The overview of a section given in the beginning of a section is not helpful at all.  Please see details in '"Any Additional Comment".

* The main contribution of the paper is not clearly presented.  A major reason is that the above presentation issues significantly hurts the readability of the paper -- even after reading the paper several times, I still cannot figure out exactly what the main contributions are.  Just take one example, the abstract says "We show that debate transforms agreement from evidence into an outcome: agreement is endogenous to the interaction that produces it."  This sentence is too vague.  Only after reading the entire paper I finally figure out that it means "We show that, while without debate the agreement among agents is evidence for accuracy, with debate that is no longer true."  Similarly vague sentences are everywhere in the paper, making the paper very difficult to read.

* What is the purpose of introducing "Calibrated Multi-Agent Debate" framework?  Do we want to mitigate the overconfience arising from multi-agent debate, or are there other purposes?  Neither the introduction nor the beginning of Section 5 explicitly say why this framework is introduced.  Let's say the purpose of this framework is to mitigate the overconfidence arising from multi-agent debate.  Then which part of the paper gives empirical evidence that this new framework can achieve that purpose?  Table 1 to Table 3?  But the text didn't explicitly say that.

* I can't understand the structure of the "Calibrated Multi-Agent Debate" framework, either.

  * This sentence at the beginning of Section 5, "_Calibrated Multi-Agent Debate (CMAD) responds with three components that trust this signal progressively less: Prevent reduces the endogeneity at the protocol level, Detect reads it from the trajectory before it saturates, and Certify replaces it with an exogenous coverage guarantee._" was meant to give an overview of the framework, but it is not helpful at all.  It only described the function of each framework component at a (too high) level, but didn't say what those components are exactly.  In my opinion, when introducing something, one should first say what it is, and then say what it does.  This paragraph only says "what it does".

  * The specific descriptions of each component of the framework, presented in Sections 5.1 to 5.4, are not helpful, either.  Let me give some examples (there are more other presentation issues):

    * Section 5.3 says that the Ceitify component "_wraps the committed answer in a distribution-free coverage guarantee under exchangeability. It requires a labeled calibration split from the deployment distribution._"  However, what is covered in "distribution-free covergae"?  What things are "exchangeable"?  What is "celibration split"?   And later in Theorem 4, what do "calibration instance" and "test instance" mean?
    * Section 5.4 says that the Affirm component "_stops at a turn t chosen on a selection fold, with the deployment threshold formed on an independent fold._"  What is a "fold"?  And later Theorem 5, what is "Bonferroni penalty"?

    As someone who are unfamilar with this area, I really cannot understand these vague sentences and undefined terms...

**Additional Comments:**

* Why is Lemma 1 called "The collapse-dynamics identity"?  Even when the dynamics do not collapse, the equations in Lemma 1 also hold, right?
* Too many terms are used before being defined or introduced without context.  For example:
  * In Definition 1 (page 4):
    * "variance analysis", which will be an important concept to be developed later in Section 4, is mentioned here without context.
    * "Under collapse, both S^2(t) and the observed disagreement vanish" should be rephrased to an explicit definition like "We define collapse as a situation where both S^2(t) and the observed disagreement vanish".  The formal definition of "collapse" is given later inside Assumption 1, which hurts readability; consider moving the definition of "collapse" to an earlier place.  Besides, "observed disagreement" is not defined.
    * "co-saturates" is not defined here.
  * What is the purpose of the sentence "the reported-confidence gap still uses the empirical co-saturation condition stated in Theorem 2" in Assumption 1?  This sentence appears without context.
  * Section 4, first paragraph: "apparent confidence $\tilde C(t)$, which the variance mechanism drives" -- the "variance mechanism" is not defined.  Also, this sentence, which describes appearent confidence and reported confidence, seems better to be presented in Section 3.
  * Section 4.1: What is an "empirical regularity"?  Do you mean "a phenomenon observed empirically"?
* Corollary 1 is confusing.  What does it want to say?  What is "usable dynamic range"?  In the sentence "For the apparent confidence this follows from Theorem 1", what does "this" refer to?  Does "terminal confidence" refer to terminal apparent confidence or terminal reported confidence?  The sentence about "ROC curve" is presented without context.  Terms like "survive" and "discriminate" are too vague to be include in a formal mathematical corollary.  What does "chance" and "near chance" mean in these two paragraphs?  Too many vague, imprecise, and unnecessary phrases obscure the main message that this corollary wants to convey.
* Section 4.2: "what remains is the one error collapse cannot average away".  Is this sentence gramatically correct?  Is it "what remains is the one (that) error collapse cannot average away" (in this case, what does "one" refer to?), "what remains is the one error (that) collapse cannot average away", or "what remains is the one error collapse (that) cannot average away"?
* Theorem 2:
  * "apparent signal" --> "apparent confidence".
  * "the local slope is XXX".  The slope of what?
  * After Theroem 2: "The identify alone is not the claim".  Then why claim this identify???
  * It seems that the sentences about monotonicity in Theorem 2 can be moved to somewhere before or inside Corollary 2.
* What do you mean by "task headroom" in Theorem 3 (Section 4.3)?
* Section 5, the first sentence: "the variance account diagnoses an endogenous trust signal".  I can't understand this sentence at all.  What is "trust signal"?  A later sentence "Only the exogenous layer is collapse-unconditional: the two signal-based components are model-dependent" -- again I can't understand.
* Section 5.2 and Corollary 1: "terminal confidence loses rank information" -- the rank of what?

**Audience:**

Yes

**Audience Explanation:**

Multi-LLM agent debate is clearly interesting to the community.

**Broader Impact Concerns:**

Positive impact, because it gives a warning of possible failure of multi-LLM debate.

**Claims And Evidence:**

No

**Claims Explanation:**

See the above Weakness part.  In particular, the theoretical result is just a restatement of assumption.  And I can't undersand the empirical framework.

**Requested Changes:**

See Additional Comments.